# 🤖OSCAR: OPERATING SYSTEM CONTROL VIA STATE-AWARE REASONING AND RE-PLANNING

**Xiaoqiang Wang**[1,2]    **Bang Liu**[1,2,3†]
[1]DIRO & Institut Courtois, Université de Montréal
[2]Mila - Quebec AI Institute    [3]Canada CIFAR AI Chair
{xiaoqiang.wang, bang.liu}@umontreal.ca

## ABSTRACT

Large language models (LLMs) and large multimodal models (LMMs) have shown great potential in automating complex tasks like web browsing and gaming. However, their ability to generalize across diverse applications remains limited, hindering broader utility. To address this challenge, we present **OSCAR**: **O**perating **S**ystem **C**ontrol via state-**A**ware reasoning and **R**e-planning. OSCAR is a generalist agent designed to autonomously navigate and interact with various desktop and mobile applications through standardized controls, such as mouse and keyboard inputs, while processing screen images to fulfill user commands. OSCAR translates human instructions into executable Python code, enabling precise control over graphical user interfaces (GUIs). To enhance stability and adaptability, OSCAR operates as a state machine, equipped with error-handling mechanisms and task-driven re-planning, allowing it to efficiently adjust to real-time feedback and exceptions. We demonstrate OSCAR's effectiveness through extensive experiments on diverse benchmarks across desktop and mobile platforms, where it transforms complex workflows into simple natural language commands, significantly boosting user productivity.

## 1 INTRODUCTION

Large Language Models (LLMs) (Ouyang et al., 2022; Achiam et al., 2023; Dubey et al., 2024) and Large Multimodal Models (LMMs) (Li et al., 2023; Team et al., 2023; Liu et al., 2024a; Reid et al., 2024) have demonstrated exceptional performance on tasks requiring complex reasoning (Liang et al., 2022; Srivastava et al., 2023; Wang et al., 2024c), particularly when combined with advanced planning techniques (Wei et al., 2022; Wang et al., 2023b;c) and external tools (Yang et al., 2023c; Liu et al., 2023a). These model-centric agents show revolutionary potential for automating real-world tasks such as web browsing (Gur et al., 2023; Deng et al., 2023; Zheng et al., 2024a), embodied simulation (Zhao et al., 2024; Shi et al., 2024), and software development (Huang et al., 2023; Hong et al., 2024a; Qian et al., 2024). However, despite impressive results, these agents struggle to generalize across different applications due to variations in observation and action spaces. In real-world scenarios, workflows often involve switching between applications and interacting with diverse graphical interfaces. This raises an intriguing and practical question: can we build a generalist agent capable of following user instructions across various applications using standardized operating system (OS) controls like mouse and keyboard inputs, while processing screen outputs?

Recent work has explored graphical user interface (GUI) control on mobile devices, with a focus on smartphone GUI understanding (You et al., 2024; Fan et al., 2024; Wu et al., 2024a) and task automation (Yang et al., 2023d; Guan et al., 2024; Zhang & Zhang, 2024; Wang et al., 2024a). For desktop computers, existing approaches simulate tasks in black-box systems like AAA games (Tan et al., 2024) and office workflows (Wang et al., 2024d). Some methods extend this to general OS control via visual question answering and human action trajectories (Hong et al., 2024b; Chen et al., 2024b; Cheng et al., 2024). However, these systems often lack real-time feedback from the OS and struggle to adapt dynamically when task execution fails. Without a grounded executable environ-

---

†Corresponding author.

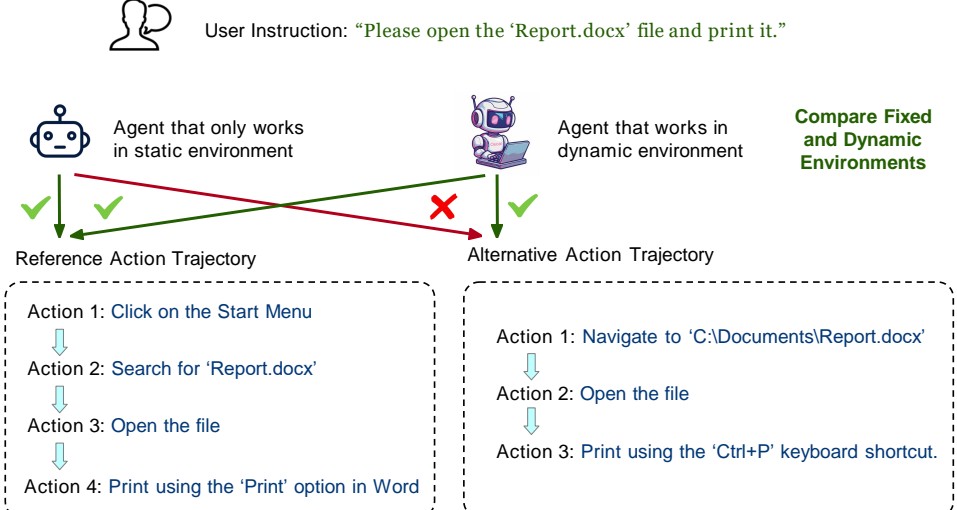

Figure 1: Comparison of agent action sequences in static and dynamic OS environments for the task of opening and printing "`Report.docx`". The static environment (left) requires a fixed action trajectory and fails if the agent deviates. The dynamic environment (right) allows for alternative action trajectories, enabling the agent to adapt and complete the task using different valid methods.

ment, these methods fall short in real-world scenarios, where real-time feedback and adaptive action adjustment are crucial for navigating new GUI environments, similar to human behavior. Recently, new executable environments (Zheng et al., 2024b; Xu et al., 2024; Xie et al., 2024b) have emerged, offering dynamic feedback and enabling agents to modify their actions on the fly, paving the way for more autonomous, adaptive agents in OS control tasks.

As depicted in Figure 1, consider an agent tasked with opening "`Report.docx`" and printing it. In a static environment, the agent must follow a predetermined sequence of actions—clicking the Start Menu, searching for "`Report.docx`", opening the file, and printing using the "`Print`" option in Word. Any deviation from this sequence, such as navigating directly to "`C:\Documents\`" directory, opening the file, and printing using the `Ctrl+P` shortcut, results in failure because the static environment cannot accept multiple valid solutions. In contrast, a dynamic environment allows the agent to adapt its actions based on real-time feedback, successfully completing the task using various valid methods. This example highlights the importance of adaptability in real-world settings, where agents must handle unforeseen changes or errors. To address this limitation, we propose leveraging a LMM to develop a generalist agent capable of interpreting user commands, interacting with graphical user interfaces (GUIs), and adjusting its strategy in response to real-time feedback.

To achieve this, we identify three key challenges in building such a generalist agent for dynamic executable environments: 1) **Unified Control Interfaces:** The agent must seamlessly operate standard input methods like mouse and keyboard across various applications. This involves executing precise actions such as mouse movements, clicks, scrolling, and using keyboard shortcuts (e.g., `Ctrl+C` for copying content), all based on visual inputs; 2) **GUI Grounding:** The agent needs to interpret the screen and accurately identify relevant elements, such as buttons, menus, or text fields. For example, when instructed to perform a web search, it must locate and interact with the search box by correctly grounding the user instructions to the on-screen components; 3) **Exploration-Based Simulation and Re-planning:** Similar to how humans navigate unfamiliar software interfaces, the agent must have the ability to explore and adjust its plan dynamically. This includes retrying actions, handling exceptions like software crashes, and adapting its strategy based on real-time feedback from the system. By addressing these challenges, we aim to develop a robust agent capable of navigating a wide range of computer applications in a flexible and reliable manner. This dynamic interaction between the agent and the operating system—driven by real-time feedback—forms the foundation of our approach, moving beyond the limitations of static, pre-scripted workflows.

In this paper, we introduce **OSCAR**, a general-purpose agent designed to autonomously interact with dynamic OS environments through code-centric control. `OSCAR` generates executable Python code to directly interface with the OS, enabling semantically clear and precise actions, ensuring broad applicability across diverse tasks. To enhance GUI understanding, `OSCAR` augment screen

observation with visual grounding and semantic grounding inputs by leveraging the OS window API to extract interactable elements and their spatial layout. `OSCAR` operates as a state machine, continuously looping through planning, action, and re-planning to handle execution failures and system exceptions. To optimize efficiency, we incorporate task-driven re-planning, allowing the agent to adjust specific tasks rather than entire workflows, minimizing overhead and enhancing adaptability in dynamic environments.

We validated `OSCAR`'s effectiveness and generalizability across diverse benchmarks involving both desktop and smartphone OS environments. On the GAIA (Mialon et al., 2023) benchmark, `OSCAR` outperformed previous methods, achieving a 28.7% average success rate, with a notable 13.5% success rate on the most complex Level 3 tasks, nearly doubling the prior state-of-the-art performance. On the OSWorld (Xie et al., 2024b) and AndroidWorld (Rawles et al., 2024) benchmarks, `OSCAR` consistently surpassed other agents, achieving a 24.5% success rate on OSWorld, and 61.6% on AndroidWorld, demonstrating superior adaptability across real-time dynamic OS tasks. These results highlight `OSCAR`'s advancement in transforming tedious tasks into natural language commands, showcasing its adaptability and strong general-purpose capability.

## 2 METHODOLOGY

In this section, we introduce `OSCAR`, an intelligent agent designed for general-purpose control and navigation within operating systems. As illustrated in Figure 2, `OSCAR` operates as a state machine (Girault et al., 1999; Yannakakis, 2000), enabling it to handle dynamic OS environments through systematic state transitions. This framework allows `OSCAR` to efficiently process user instructions, observe the environment, plan and execute actions, and verify outcomes, while managing potential OS exceptions. We now detail the state transition process, highlighting how `OSCAR` integrates GUI grounding, task-driven re-planning, and code-centric control in each operational state.

### 2.1 FORMULATION OF STATE TRANSITIONS

**[Init → Observe].** In the `[Init]` state, `OSCAR` awaits user instructions. Upon receiving a command, the system transitions to the `[Observe]` state to begin processing the input. This is the starting point for each task, and the agent returns to this state after completing or terminating a task.

**[Observe → Plan].** After receiving the user's request, `OSCAR` captures a screenshot of the current environment and interprets it by performing GUI grounding detailed in Section 2.2. This involves identifying screen elements, such as buttons and input fields, to understand the user interface context. The system then transitions to the `[Plan]` state.

**[Plan → Execute, Plan → Verify].** In the `[Plan]` state, `OSCAR` generates an action plan based on the current screenshot, user instructions, context memory, and any previous verification feedback from the OS (if available). As detailed in Section 2.3, it utilizes task-driven re-planning to invoke the model backend and determine the next action.

- If more actions are needed to complete the task, `OSCAR` stores the planning results and generated actions in the context memory and transitions to the `[Execute]` state to interact with the operating system via executable Python code.
- If no further actions are necessary (the whole task completion is indicated), `OSCAR` transitions directly to the `[Verify]` state.

**[Execute → Plan, Execute → Observe → Plan].** In the `[Execute]` state, the Python code is executed to interact with the operating system. There are two possible outcomes:

- If execution fails due to invalid code (*e.g.*, attempts to access non-existent GUI elements), `OSCAR` transitions back to the `[Plan]` state, incorporating the interpreter's error message for re-planning.
- If execution succeeds, `OSCAR` first transitions to the `[Observe]` state to capture a new screenshot, reflecting the updated state of the environment. Subsequently, `OSCAR` moves to the `[Plan]` state to plan the next action based on the new context.

**[Verify → Success, Verify → Plan, Verify → Fail].** In the `[Verify]` state, `OSCAR` runs evaluation scripts to validate the outcomes of the executed actions. These scripts

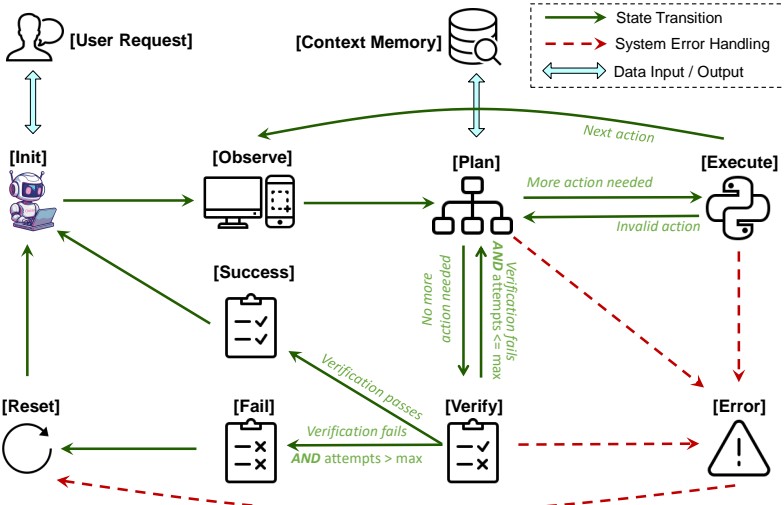

Figure 2: Illustration of the state machine model used in `OSCAR`. The model consists of multiple states—`[Init]`, `[Observe]`, `[Plan]`, `[Execute]`, `[Verify]`, `[Success]`, `[Fail]`, `[Reset]`, and `[Error]`—and handles transitions between them. Transitions are triggered by user request, planning completion, verification results, or OS errors.

check system or application settings and analyze file content to confirm that the intended tasks were successfully completed. Based on the results, `OSCAR` either transitions to the `[Success]` state if verification passes or returns to the `[Plan]` state if it fails. If the failure exceeds the allowed maximum number of attempts, `OSCAR` transitions to the `[Fail]` state.

**[Success → Init].** If the task verification passes, `OSCAR` enters the `[Success]` state, signaling successful task completion and notifying the user. The system then transitions to the `[Init]` state, ready to process the next user query.

**[Fail → Reset].** If the task cannot be completed after the maximum number of allowed attempts, `OSCAR` transitions to the `[Fail]` state, notifying the user of the failure and then transitioning to the `[Reset]` state.

**[Plan → Error, Execute → Error, Verify → Error, Error → Reset].** `OSCAR` transitions to the `[Error]` state when a critical system exception or crash occurs, such as a local model backend failure or when too many files or processes are open in the OS. In this state, the task is terminated, and the user is notified of the error. User intervention may be required to resolve the issue before `OSCAR` transitions to the `[Reset]` state.

**[Reset → Init].** In the `[Reset]` state, `OSCAR` restores the operating system to its pre-query configuration by terminating processes and closing file handlers. Once the reset is complete, `OSCAR` returns to the `[Init]` state, ready to process the next user query.

In a nutshell, the state machine architecture of `OSCAR` introduces continuous feedback loops, enabling dynamic interaction and error recovery, which enhances its robustness in dynamic OS environments. Additionally, unlike previous methods that relied on linear action sequences and re-planning from scratch (Yang et al., 2023d; Zhang et al., 2024a; Wu et al., 2024c), `OSCAR`'s state machine integrates real-time verification feedback for fine-grained, task-driven re-planning, significantly improving efficiency and adaptability. Most importantly, its modular state transitions allow for flexible generalization across diverse OS environments, such as desktop and smartphone OS.

## 2.2 GUI-GROUNDED OBSERVATION

While LLMs exhibit strong capabilities in understanding general visual information and grounding in broad domains, feeding a screenshot into the model to facilitate planning and output control over the screen remains insufficient. This insufficiency stems from the fact that GUI images differ significantly from natural images (Cheng et al., 2024), as they are densely packed with text and

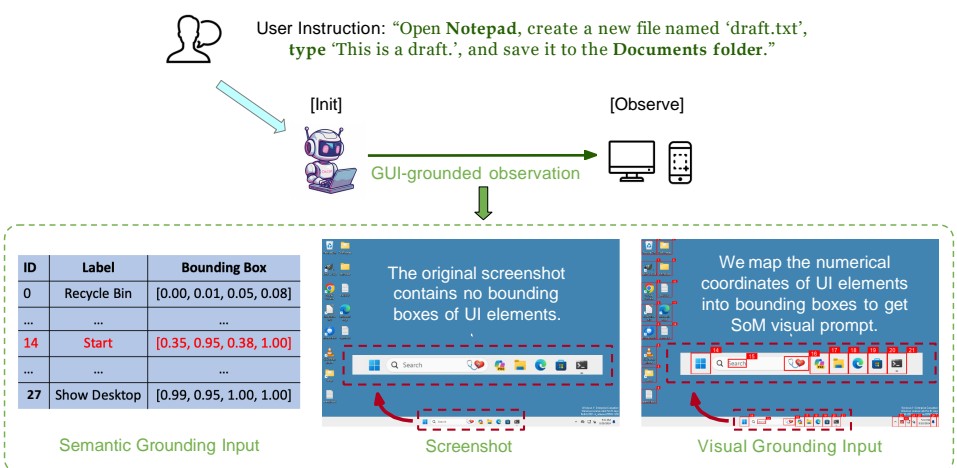

Figure 3: Illustration of GUI-grounded observation in OSCAR, which includes original screenshot, semantic grounding input, and set-of-mark (SoM)-based visual grounding input.

diverse interaction elements, such as icons and widgets, often rendered at a small scale relative to high-resolution screens. As a result, it is difficult for models to accurately locate all interaction elements and understand GUI semantics. For instance, both ⚙ and ⚞ could represent a settings icon, depending on the application.

To address this, we introduce a dual-grounding observation approach to enhance GUI understanding, *i.e.* incorporating both visual grounding and explicit semantic grounding. Firstly, we leverage a Set-of-Mark (SoM) prompting (Yang et al., 2023a) technique to enhance GUI visual grounding. SoM prompting, a visual prompting technique that adds marks to image regions to significantly improve LMM performance on fine-grained vision tasks. Specifically, we utilize native window API to extract the Accessibility (A11Y) tree, a kind of structural representation providing the location, properties, and states of UI components (Consortium, 2018). Based on the A11Y tree, we extract precise numerical coordinates of UI elements and map them into bounding boxes to generate SoM visual prompts (Figure 3). The A11Y tree offers greater precision and robustness than the commonly adopted detection+OCR pipeline (Gao et al., 2023; Wang et al., 2024a), particularly in complex screens with numerous UI elements where OCR often fails (see Section 3.1 for ablation analysis).

In addition to visual grounding, we further enhance GUI understanding through explicit semantic grounding by adding descriptive labels to key elements, such as: (ID: 14, Label: Start, $X_1$: 0.35, $Y_1$: 0.95, $X_2$: 0.38, $Y_2$: 1.00). These labels not only offer semantic descriptions of UI components but also facilitate code-centric control by allowing precise references to elements (*e.g.* by element ID).

By combining the screenshot with dual-grounding observations, OSCAR can not only grasp the overall layout and context of the GUI, but also focus on relevant areas of the screen, while flexibly referring to specific elements when needed. This approach significantly enhances GUI understanding, ensuring robust and efficient task execution in dynamic OS environments.

## 2.3 TASK-DRIVEN RE-PLANNING

Interacting with dynamic environments for open-ended tasks has been well-studied in domain-specific agents, such as those agents in Minecraft (Wang et al., 2023a;d) and data analysis (Guo et al., 2024; Zhang et al., 2024c). Iterative planning with exploration in self-instructed task curricula has proven effective, as agents adjust their plans based on environmental feedback. These methods typically involve two stages: *exploration* and *deployment*. During the exploration phase, agents comprehensively interact with the environment to gather knowledge and experience. In the deployment phase, agents apply the learned strategies from exploration to operate and navigate new environments.

However, while navigating dynamic operating systems shares the goal of determining feasible action sequences for complex tasks, it introduces significant efficiency challenges, as agents must respond

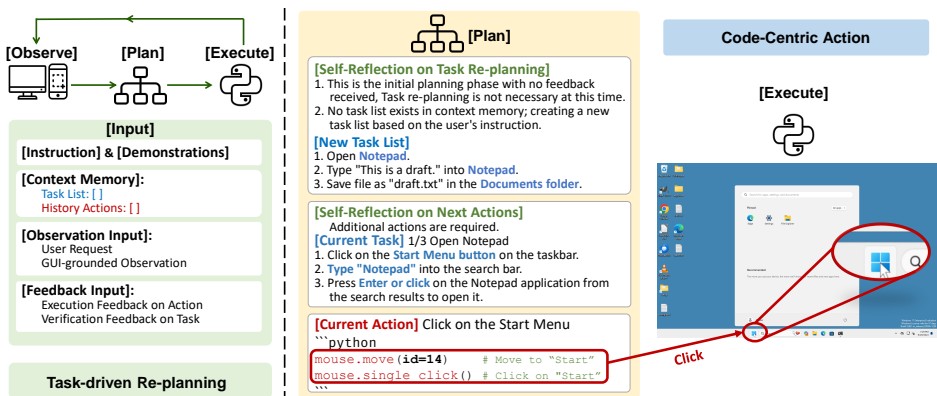

Figure 4: Illustration of task-driven re-planning and code-centric control in OSCAR. Based on the current observation, context memory, and real-time OS feedback from execution or verification, OSCAR generates a refined task list and determines the next action. The action refers to GUI elements using semantic grounding input and includes executable Python code to control the OS, such as clicking the "Start" button (id=14 in Figure 3) and launching applications.

promptly to user requests. The plan-after-fully-exploration approach is inefficient for OSCAR in these contexts. To balance efficiency and effectiveness, we introduce **task-driven re-planning**, while storing action trajectories and planning results in context memory to summarize and leverage past experiences. Specifically, we draw inspiration from plan-and-solve prompting (Wang et al., 2023b; Zhang et al., 2024b), a planning-based chain-of-thought (Wei et al., 2022) approach that simplifies complex tasks by breaking them into a hierarchy of sub-tasks and mapping them into executable actions. As shown in Figure 4, we instantiate this concept as two-level planning. Level 1: Decompose user instructions into tasks using standardized operating procedures (SOPs) (Hong et al., 2024a), improving clarity in task decomposition. Level 2: For each task, generate actions step-by-step, interleaving planning and execution within OSCAR's state machine.

A significant advantage of task-driven re-planning is fine-grained self-refinement (Shinn et al., 2023; Tao et al., 2024), *i.e.* when negative feedback is received from dynamic evaluation in the state transition of [Verify] → [Plan], OSCAR can re-plan only specific tasks, rather than re-planning the entire workflow or just the current action. This approach improves planning efficiency by enabling fine-grained re-planning of tasks. It also helps avoid error propagation (Zhang & Zhang, 2024), where incorrect actions early on prevent successful completion of user requests, regardless of how well subsequent actions are planned. For example, in a workflow involving multiple applications—extracting information from a Word document, observing a figure in Photos, and summarizing content in PowerPoint—each task requires several interactions. Errors in earlier tasks, such as copying text or capturing an image, will propagate and result in incorrect summaries in PowerPoint.

Formally, the complete prompt input for invoking the model is summarized in Figure 4, which includes user request, context memory, GUI-grounded observation and feedback from both execution and verification phases. The full version of system prompt can be found in the Appendix B.

## 2.4 CODE-BASED ACTION

As portrayed in Figure 4, leveraging the textualized SoM from observed screenshots, OSCAR can easily refer interaction elements on the screen using element ID or numerical coordinates. This allows OSCAR to generate code to control these elements with logically clear semantics. To operationalize OSCAR's action space, we employ the widely-used PyAutoGUI library [1] for mouse and keyboard control. This library enables various mouse behaviors (movement, click, scroll) and keyboard interactions (single key presses, key shortcuts). Further details are summarized in Table 5.

---

[1]https://pyautogui.readthedocs.io/

## 3 Experiments

**Benchmarks.** We evaluate `OSCAR` on real-world workflow automation benchmarks involving complex user requests. The first benchmark is GAIA (Mialon et al., 2023), which consists of 466 question-answering (QA) structured into three levels: Level 1 includes simple tasks requiring no more than five steps; Level 2 involves more complex tasks with 5-10 steps and multiple tools; and Level 3 presents advanced tasks requiring over 10 actions and tool usage. The second benchmark is OSWorld (Xie et al., 2024b), an interactive dynamic environment with real-time OS feedback. It includes 369 tasks covering OS settings, office software, daily applications (*e.g.* Chrome), professional tools (*e.g.* VSCode), and multi-application tasks. Without a gold-standard reference action sequence, the environment allows for multiple valid solutions, which are evaluated through dynamic execution testing—verifying modified files or displayed text content in windows. Additionally, similar to OSWorld, AndroidWorld (Rawles et al., 2024) provides a dynamic smartphone OS environment with 116 tasks spread across 20 diverse applications, and human annotated difficulty level: easy, medium, hard. *Please refer to Appendix D and Appendix E for more experiments on the GUI understanding and static navigation benchmark.*

**Baselines.** We compare `OSCAR` with seven agents designed to handle dynamic OS feedback. For the desktop OS environment, we include Cradle (Tan et al., 2024), UFO(Zhang et al., 2024a), FRIDAY (Wu et al., 2024c), and MMAC (Song et al., 2024). For the smartphone OS environment, we evaluate against M3A (Rawles et al., 2024), AppAgent (Yang et al., 2023d), and Mobile Agent (Wang et al., 2024a). Implementation details of `OSCAR` and these baselines are provided in Appendix B.

**Results.** Table 1 summarizes the results on the GAIA benchmark, where `OSCAR` achieves the best performance across all three levels of workflow complexity. In particular, for Level 3 tasks, `OSCAR` significantly outperforms previous methods, achieving 13.5% compared to MMAC's 6.1%, demonstrating the effectiveness of `OSCAR`'s task-based planning. Additionally, as shown in Table 2, `OSCAR` consistently surpasses other methods across various applications in dynamic desktop OS environments. In challenging tasks involving multiple applications, `OSCAR` achieves a 12.9% success rate, outperforming the multi-agent baseline, UFO, which leverages dual agents to coordinate workflow decomposition and execution. When adapting `OSCAR`'s action space to a mobile environment, as shown in Table 3, it achieves better average performance than the two-phase approach (comprehensive exploration followed by execution) of AppAgent, particularly in the medium and hard subsets, highlighting the effectiveness and efficiency of `OSCAR`'s task-driven re-planning.

Table 1: Real-world workflow results on the GAIA benchmark using the exact match metric. Since MMAC does not publicly release their code, we report MMAC's results as stated in their paper and use the same base model (*i.e.* GPT-4-turbo ) in all of the baseline models, for a fair comparison.

| Model | Level 1 | Level 2 | Level 3 | Average |
|---|---|---|---|---|
| GPT-4-turbo | 9.7 | 6.9 | 0.0 | 5.5 |
| GPT-4 plugins | 30.3 | 9.7 | 0.0 | 13.3 |
| UFO | 36.9 | 16.1 | 5.4 | 19.4 |
| FRIDAY | 40.9 | 20.1 | 6.1 | 22.4 |
| MMAC | 45.2 | 20.8 | 6.1 | 24.0 |
| **OSCAR** | **47.0** | **25.6** | **13.5** | **28.7** |

Table 2: Quantitative results on the OSWorld benchmark, measured by success rate (SR). All baselines incorporate the SoM visual prompt as auxiliary GUI-grounded input and use GPT-4o as the base model to ensure a fair comparison.

| Model | OS | Office | Daily | Prof. | Multi | Avg. |
|---|---|---|---|---|---|---|
| Cradle | 16.7 | 3.5 | 6.6 | 20.4 | 5.5 | 10.5 |
| UFO | 37.5 | 6.8 | 12.8 | 14.3 | 10.9 | 16.5 |
| FRIDAY | 45.8 | 8.5 | 14.1 | 18.4 | 6.9 | 18.8 |
| **OSCAR** | **58.3** | **12.0** | **16.7** | **22.4** | **12.9** | **24.5** |

Table 3: Quantitative results on the AndroidWorld benchmark using the same model and input settings as OSWorld.

| Model | Easy | Medium | Hard | Average |
|---|---|---|---|---|
| M3A | 41.0 | 33.3 | 26.3 | 33.5 |
| Mobile Agent | 49.2 | 41.7 | 31.6 | 40.8 |
| AppAgent | **82.0** | 55.6 | 42.1 | 59.9 |
| **OSCAR** | 65.6 | **66.7** | **52.6** | **61.6** |

### 3.1 Ablation analysis

We conduct ablation analysis on the individual components of `OSCAR`, including GUI-grounded observation and various planning techniques. Specifically, we first compare our GUI-grounded observation against baseline that omits the dual-

grounding input, *i.e.* feeding raw screenshots as input. Additionally, we replace A11Y tree-based extraction with a Detection+OCR pipeline.

For the baselines in planning techniques, we replace our task-driven re-planning with state-of-the-art methods used in multi-step decision-making tasks, particularly for long action sequences. These include ReAct (Yao et al., 2022b), plan-and-solve (Wang et al., 2023b), and chain-of-action (Zhang & Zhang, 2024).

The results of different baselines on the OS-World benchmark are illustrated in Figure 5. We have the following observation: 1) Both GUI-grounding and task-driven re-planning significantly enhance performance. Specifically, raw screen input without GUI grounding and direct prompts without fine-grained planning achieve only 70% and 80% of OSCAR's full performance, respectively. 2) The Detec-

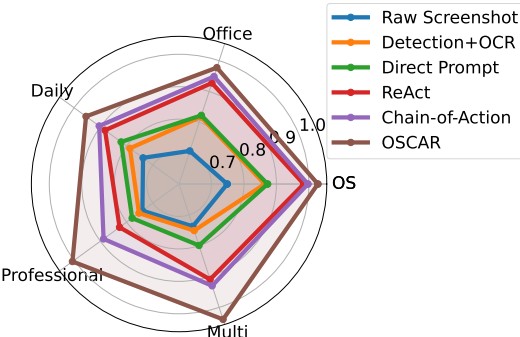

Figure 5: Decomposed performance of various baselines, with scores normalized using max-min scaling for each capability to enhance clarity.

tion+OCR pipeline is less effective than the original A11Y tree-based method, particularly on the subset of professional tools with numerous UI elements, where it only marginally outperforms raw screenshot input. Furthermore, the Detection+OCR method introduces additional processing time, reinforcing the A11Y tree as the superior choice for dynamic OS environments. 3) Advanced planning strategies can significantly enhance workflow performance. For instance, ReAct and Chain-of-Action achieve results that are comparable to OSCAR in daily application and office software scenarios. 4) Without considering real-time OS feedback and efficient re-planning, ReAct and Chain-of-Action struggle in professional software and multi-application scenarios, highlighting OSCAR's advantage in adapting to dynamic OS environments.

## 3.2 IN-DEPTH ANALYSIS

**Instance-level analysis on planning efficiency.** To better understand why OSCAR achieves superior performance, particularly in dynamic OS environments, we take a closer look at the final success rate results and conduct an instance-level analysis for both successful and failed user requests on the OSWorld benchmark. Specifically, for the successful cases with OSCAR, we track the number of re-planning occurrences before verification failures exceed the allowed maximum number of attempts *i.e.* the upper bound for re-planning is the maximum number of allowed attempts. We also track the total action steps taken and the ratio of the successful action path length to the total steps, serving as a proxy for the action matching score in dynamic environments, where no reference action path exists as it does in static environments (Rawles et al., 2023). It is used to quantify the planning and execution efficiency in the fail-and-re-planning setting, is also referred as process score (PS) by Wang et al. (2024a), or as completion rate (CR) by Zhang et al. (2024a).

For failed cases, following Xu et al. (2024), we categorize failures into three classes: False Completion (FC), where the agent incorrectly believes the task is completed; Reach Step Limit (RSL), where the agent reaches the maximum step limit without completing the task; and Invalid Action (IA), where the agent produces outputs that do not follow instructions, including invalid formats, nonexistent actions, or incorrect action parameters. Since OSCAR can handle invalid actions and false completions through execution and verification feedback, *i.e.* [Execute] → [Plan] and [Verify] → [Plan] state transitions, FC and IA errors do not occur in OSCAR. We further analyze a subclass of RSL, where re-planning generates the same task list or action trajectory that has already been marked as a verification failure in previous attempts. We refer to this subclass as Redundant Re-plan (RR). For comparison, we also analyze these metrics for FRIDAY, the most competitive baseline in dynamic OS environments, as shown in Table 2.

**OSCAR requires fewer re-planning attempts.** As shown in Figure 6, in the successful requests, over 80% of the samples using OSCAR required fewer than 3 re-planning attempts, whereas in FRIDAY, more than 50% of the successful samples needed 3 to 4 re-planning attempts

(the maximum allowed re-planning attempts in our experiments is 4, after which the case is deemed a failure). This distribution highlights OSCAR's efficiency advantages, as it leverages task-driven re-planning to focus on high-level task lists and perform fine-grained adjustments, rather than re-planning the entire workflow. These findings align with our goal of adapting to dynamic OS feedback while improving efficiency.

**OSCAR's re-planning includes smaller, more efficient steps.** The proxy action matching score indicates that OSCAR consistently takes smaller, more efficient steps during re-planning, while FRIDAY's score worsens as the number of re-planning attempts increases. This efficiency is due to OSCAR's ability to learn from previous trials, using the stored task list and action history in its context memory to optimize subsequent task lists and action trajectories upon receiving verification failure feedback.

**OSCAR's failure cases involve less redundant re-planning.** As shown in Table 4, while OSCAR may not always complete the user request within the allowed attempts, its re-planning effectively avoids repeating previous steps. In contrast, FRIDAY's tendency to re-plan the entire workflow frequently (52.8%) results in generating an action trajectory that has already been verified as unsuccessful. This finding complements the success case results, where most of OSCAR's successful cases required only 1-2 re-planning attempts.

**Qualitative examples.** As illustrated in Figure 7, OSCAR effectively handles complex requests involving multiple applications, *i.e.* OS→Office→OS→Daily, showcasing its flexible and effective planning. Please refer to Appendix F for more qualitative examples.

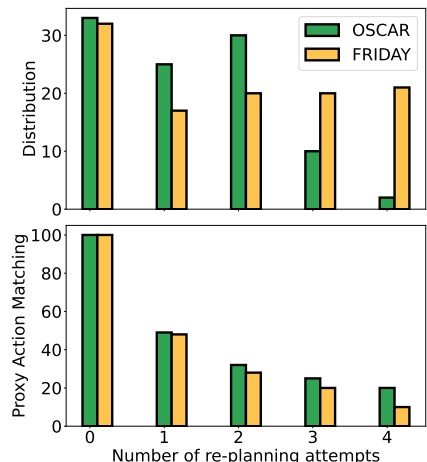

Figure 6: Planning efficiency analysis of successful cases.

Table 4: Failure case statistics for False Completion (FC), Reach Step Limit (RSL), and Invalid Action (IA). The subclass of RSL, Redundant Re-plan (RR), is also reported as a ratio relative to the total number of failure cases.

| Model | FC | RSL (RR) | IA |
|-------|------|--------------|-------|
| FRIDAY | 9.1% | 70.4% (52.8%) | 20.5% |
| OSCAR | – | 100% (15.2%) | – |

## 4 RELATED WORKS

**GUI agents.** LLM and LMM-based agents (Wang et al., 2024b; Xie et al., 2024a; Madaan et al., 2023) have been developed across various environments, including robotics (Driess et al., 2023; Zitkovich et al., 2023), web browsing (Yao et al., 2022a; Gur et al., 2023), gaming (Fan et al., 2022), software development (Yang et al., 2023b), automating benchmark construction (Liu et al., 2024b), data analysis (Zhang et al., 2024c), and AI for science (Xiao et al., 2024). Among them, GUI agents—capable of interacting with various desktop and smartphone GUIs—offer broader applicability in automating real-world workflows (Mialon et al., 2023). Some agents are continually pre-trained (Cheng et al., 2024) or fine-tuned (Chen et al., 2024b) on GUI-specific data. Others simulate GUI control in sandbox environments, such as AAA games (Tan et al., 2024) or office workflows (Wang et al., 2024d), which require internal application-specific APIs to interact with the environment. In a broader context, some agents interact with basic OS APIs but are often designed for static, pre-defined environments (Reed et al., 2022; Hong et al., 2024b) without grounding in real-time executable environments. Other agents follow linear action sequences and perform re-planning from scratch (Yang et al., 2023d; Zhang et al., 2024a; Wu et al., 2024c) when verification fails, lacking fine-grained re-planning strategies, which makes them less efficient in real-world scenarios. Motivated by these limitations, we design OSCAR to handle real-time dynamic OS feedback using an efficient, state-aware, task-driven re-planning strategy.

**Synergizing LLMs and LMMs with OS.** Beyond GUI agents, another line of work explores integrating LLMs and LMMs with OSs in two key areas: 1) optimizing or tuning traditional OS functions using LLMs, and 2) integrating LLMs into OS kernels (LLM as OS) to serve as system-level interfaces, facilitating local agent operations and deployment. The former includes optimizing CPU load balancing (Li et al., 2024), improving storage access (Wu et al., 2024b), and identifying

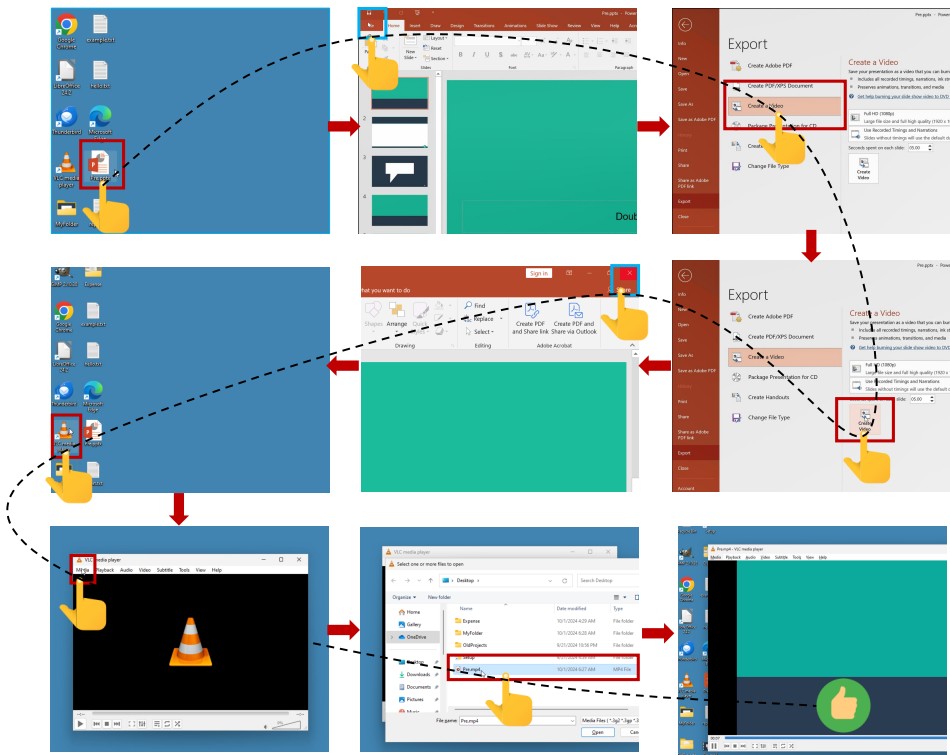

Figure 7: Qualitative results when processing user request "*Could you please convert 'Pre.pptx' to video and play it with VLC?*" on the OSWorld benchmark. Some intermediate steps and other regions of the screenshot have been omitted for clarity.

and repairing code vulnerabilities (Islam et al., 2024). The latter focuses on OS-level hardware adaptation and resource management (Kamath & Yadalam, 2024) as well as agent-level resource scheduling and sharing (Mei et al., 2024; Zhuo et al., 2024; Yang et al., 2024), such as managing agent memory and enabling efficient communication among multiple heterogeneous agents sharing the same model back-end. Unlike these approaches, OSCAR functions as a generalist GUI agent, acting as an OS co-pilot to enhance user experience and productivity.

## 5 CONCLUSION

In this work, we introduced OSCAR, a generalist agent that autonomously navigates and interacts with dynamic OS environments using a code-centric control framework. By leveraging task-driven re-planning and GUI-grounded observations, OSCAR achieves robust adaptability and effectiveness across both desktop and smartphone OS tasks. Our experiments on real-world workflow automation benchmarks, including GAIA, OSWorld, and AndroidWorld, demonstrate significant improvements in task success rates, particularly for complex, multi-step workflows.

Despite its strengths, OSCAR faces challenges in safety and lifelong self-improvement. Reliable metrics for assessing agent safety and detecting latent side effects are lacking, and the [Verification] state focuses on task accuracy without fully addressing harmful actions. Additionally, while task-level adaptation is effective, cross-task learning remains constrained by scalability challenges and risks of overfitting.

To address these issues, we plan to integrate action and tool filters for critical operations and introduce safeguards such as requiring human confirmation for actions with real-world consequences. For self-improvement, we aim to explore structured state representations beyond pre-defined transitions, investigate a dual-agent framework for adaptive state management, and enhance memory summarization and retrieval for large-scale task deployments. These advancements will further improve OSCAR's safety and adaptability in complex environments.

## ACKNOWLEDGEMENTS

This work is supported by the Canada CIFAR AI Chair Program and the Canada NSERC Discovery Grant (RGPIN-2021-03115).

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

## A    OVERVIEW

In the Appendix, we present:

- Implementation details in Appendix B.
- Baseline details in Appendix C.
- Experiments on GUI understanding benchmarks in Appendix D.
- Experiments on static GUI navigation benchmarks in Appendix E.
- Additional qualitative results in Appendix F.

## B    IMPLEMENTATIONS

**Observation space.** In dynamic OS environments, we extract Set-of-Mark (SoM) using native system APIs to obtain the Accessibility (A11Y) tree, as described in Section 2.2. For ablation study in Section 3.1 and other benchmarks without a dynamic OS environment, as described in Appendix D and Appendix E, *i.e.* only providing a screenshot, we employ an Detection+OCR pipeline to extract SoM. Specifically, we follow Gao et al. (2023); Wang et al. (2024a) and use YOLO-v8 (Reis et al., 2023) and Google OCR (Google Cloud, 2024) to parse the GUI into SoM visual prompts, serving as auxiliary inputs for screen observation.

**Action space.** The action space of OSCAR in desktop OS and smartphone OS is summarized in Table 5. This action space is used in the dynamic OS environments, *i.e.* OSWorld (Xie et al., 2024b) and AndroidWorld (Rawles et al., 2024). For the GUI understanding benchmark described in Appendix D and the static GUI navigation benchmark in Appendix E, we adapt the action space to meet the benchmark requirements, *i.e.* free-form answering text format in the GUI understanding benchmark, and structural output including predefined action types and selected elements or location coordinates.

**Base model.** To ensure a fair comparison, we set the base model of OSCAR and all baseline models to GPT-4o, *i.e.* `gpt-4o-2024-05-13`, except for the results on GAIA in Table 1, which are based on GPT-4-turbo, *i.e.* `gpt-4-turbo-2024-04-09`, since the baseline MMAC (Song et al., 2024) does not publicly release their code and their results are based on GPT-4-turbo. The temperature of response generation is set to 0.1 to reduce the variance in text generation. We provide 8 in-context demonstration examples to help the model better understand the instruction. These examples do not include a screenshot but provide a description of the current screen. All baselines are also provided with 8 in-context demonstrations to ensure a fair comparison. The full version of system prompt are provided in Figure 10 and Figure 11.

**Experiment setup.** We conduct evaluation experiments on 2 A100 GPUs. Since fine-tuning the base model is not involved and it is accessed via API, the GPU is mainly required for the Detection+OCR pipeline. As this pipeline is efficient on CPU machines, all experiments can also run on regular Windows 11 machines with WSL virtualization support, which is used for encapsulating the development and test environments in Docker containers. The maximum number of allowed attempts per run is set to 4. We report the average results across 4 runs for each model on each benchmark.

## C    BASELINE DETAILS

We employ four types of baselines for a comprehensive and fair comparison with OSCAR, categorized along two orthogonal dimensions: 1) whether the baseline is based on general-purpose out-of-the-box LMMs, or specialized LMMs that have been continually pre-trained (without human annotations) or fine-tuned (with curated human annotations) on GUI-specialized data, and 2) the target GUI scenario, whether the agent is developed for desktop OS or smartphone OS. These baselines are summarized in Table 6. To the best of our knowledge, OSCAR is the first agent capable of navigating both desktop and smartphone OS environments while responding to real-time OS feedback.

Table 5: The formulation of action space of `OSCAR` to navigate in desktop OS (top part) and smartphone OS (bottom part).

| Action | Parameter | Description |
|---|---|---|
| move | id: int | Move the mouse cursor to the GUI element labeled with id |
| | (x: float, y: float) | Move the mouse cursor to given coordinate (x, y) |
| single_click | − | Click the left button of mouse at current position |
| double_click | − | Click the left button twice of mouse at current position |
| right_click | − | Click the right button of mouse at current position |
| scroll | dist: int | Scroll the mouse wheel with distance dist |
| drag | (x₁: int, y₁: int, x₂: int, y₂: int) | Hold and drag the mouse cursor from $(x_1, y_1)$ to $(x_2, y_2)$ |
| press | key: str | Press given key or keyboard shortcuts in current window |
| write | text: str | Write down the given text in current window |
| tap | id: int | Tap on the GUI element labeled with id |
| | (x: float, y: float) | Tap the screen on given coordinate (x, y) |
| long_tap | id: int | Press and hold the GUI element labeled with id |
| | (x: float, y: float) | Press and hold screen on given coordinate (x, y) |
| swipe | (id: int, dir: str, dist: float) | Swipe on an element labeled with id in a given direction dir (up, down, left, right) and distance dist. |
| swipe | (x: int, y: int, dir: str, dist: float) | Swipe from the coordinate (x, y) on the screen in a given direction dir (up, down, left, right) and distance dist. |
| write | text: str | Write down the given text in current text field |

Table 6: Baselines for comparison with `OSCAR`, categorized by general-purpose out-of-the-box (OOTB) vs. specialized fine-tuned (FT) base LMMs and their target GUI environment (desktop or smartphone OS). `OSCAR` uniquely navigates both environments with real-time OS feedback.

| Agent | Base Model | Desktop OS | Smartphone OS | Dynamic Feedback |
|---|---|---|---|---|
| Auto-GUI (Zhang & Zhang, 2024) | OOTB | ✗ | ✓ | ✗ |
| SeeAct (Zheng et al., 2024a) | OOTB | ✓ | ✗ | ✗ |
| CogAgent (Hong et al., 2024b) | FT | ✓ | ✓ | ✗ |
| SeeClick (Cheng et al., 2024) | FT | ✓ | ✓ | ✗ |
| GUICourse (Chen et al., 2024b) | FT | ✓ | ✓ | ✗ |
| AppAgent (Yang et al., 2023d) | OOTB | ✗ | ✓ | ✓ |
| Mobile Agent (Wang et al., 2024a) | OOTB | ✗ | ✓ | ✓ |
| M3A (Rawles et al., 2024) | OOTB | ✗ | ✓ | ✓ |
| WebAgent (Gur et al., 2023) | FT | ✓ | ✗ | ✓ |
| FRIDAY (Wu et al., 2024c) | OOTB | ✓ | ✗ | ✓ |
| UFO (Zhang et al., 2024a) | OOTB | ✓ | ✗ | ✓ |
| MMAC-Copilot (Song et al., 2024) | OOTB | ✓ | ✗ | ✓ |
| Cradle (Tan et al., 2024) | OOTB | ✓ | ✗ | ✓ |
| OSCAR | OOTB | ✓ | ✓ | ✓ |

## D GUI UNDERSTANDING BENCHMARK

**Benchmarks and Evaluation.** To testify `OSCAR` whether possess a robust understanding of various GUI scenarios, including different OS platform and multi-window interactions, we firstly evaluation `OSCAR` on a comprehensive GUI understanding benchmark - GUI-World(Chen et al., 2024a). GUI-World covering six GUI scenarios across Desktop OS and Smartphone OS and formulated as a visual question-answering task. Specifically, Given one or multiple screenshots, the agent outputs a summarized caption, layout description, and GUI elements, or infers relations between screenshots. Following Chen et al. (2024a), we evaluate performance using automatic metrics for natural language generation, such as BERTScore(Zhang et al., 2019) and LLM-as-a-Judge methodology (Liu et al., 2023b; Zheng et al., 2023), or accuracy metric for multiple-choice questions.

**Results.** As shown in Table 7, we observe that: 1) `OSCAR` achieves the best GUI understanding performance across five types of GUI domains, except for websites, where the state-of-the-art agent uses an advanced parser to extract HTML as input. When HTML text is provided to `OSCAR` as additional input, it also demonstrates state-of-the-art performance in website GUI understanding. This success can be attributed to `OSCAR`'s GUI-grounded observation, which we further analyze in Section 3.2. 2) Fine-tuning on domain-specific data slightly compromises performance in more general domains. For example, the Web Agent achieves 83 on iOS GUI, significantly lower than its

Table 7: Quantitative results on the GUI-World benchmark covering six types of GUI domains.

| Model | Software | | Website | | XR | | Multi | | iOS | | Android | |
|---|---|---|---|---|---|---|---|---|---|---|---|---|
| | MC | Free | MC | Free | MC | Free | MC | Free | MC | Free | MC | Free |
| SeeAct | 93.9 | 4.328 | 91.1 | 4.167 | 90.6 | 4.031 | 90.1 | 4.172 | 84.8 | 3.750 | 92.3 | 3.865 |
| Auto-GUI | 94.8 | 4.422 | 90.5 | 4.131 | 89.0 | 3.904 | 88.0 | 4.073 | 84.0 | 3.666 | 91.4 | 3.742 |
| CogAgent | 94.4 | 4.322 | 87.5 | 3.976 | 90.1 | 4.031 | 88.3 | 4.086 | 88.7 | 4.193 | 93.6 | 4.056 |
| SeeClick | 91.0 | 4.083 | 88.5 | 4.038 | 89.5 | 3.893 | 89.7 | 4.176 | 88.2 | 4.078 | 94.3 | 4.124 |
| GUICourse | 92.4 | 4.156 | 88.9 | 4.038 | 90.3 | 4.057 | 88.6 | 4.111 | 88.4 | 4.168 | 92.9 | 4.058 |
| AppAgent | 86.5 | 3.644 | 84.3 | 3.805 | 90.8 | 4.159 | 89.5 | 4.176 | 90.6 | 4.398 | 95.0 | 4.326 |
| Mobile Agent | 88.6 | 3.822 | 86.0 | 3.877 | 90.6 | 4.047 | 89.7 | 4.199 | 91.1 | 4.482 | 94.9 | 4.230 |
| M3A | 88.1 | 3.799 | 84.1 | 3.803 | 92.1 | 4.270 | 94.1 | 4.456 | 89.6 | 4.278 | 93.5 | 4.127 |
| WebAgent | - | - | - | - | - | - | - | - | - | - | - | - |
| FRIDAY | 95.1 | 4.406 | 89.4 | 4.090 | 87.7 | 3.662 | 86.5 | 3.991 | 85.2 | 3.768 | 92.0 | 3.845 |
| UFO | 94.4 | 4.352 | 91.0 | 4.182 | 91.4 | 4.159 | 89.8 | 4.179 | 84.8 | 3.778 | 90.6 | 3.649 |
| MMAC | - | - | - | - | - | - | - | - | - | - | - | - |
| OSCAR | 96.4 | 4.509 | 89.2 | 4.035 | 94.4 | 4.551 | 95.5 | 4.527 | 92.7 | 4.585 | 96.4 | 4.524 |
| OSCAR+HTML | - | - | 92.2 | 4.235 | - | - | - | - | - | - | - | - |

state-of-the-art performance of 93 on website GUI. 3) The average performance difference among the agents is marginal, highlighting the strong single-step GUI understanding capability of the base model, GPT-4o, used in our experiments.

## E STATIC GUI NAVIGATION BENCHMARK

**Benchmarks and Evaluation.** We evaluate OSCAR on GUI navigation benchmarks involving multi-step decision-making in pre-defined interaction episodes, which includes widely adopted datasets such as Mind2Web(Deng et al., 2023) (Desktop OS) and AITW(Rawles et al., 2023) (Smartphone OS). These benchmarks consist of high-level task descriptions, gold reference sequences of actions, and corresponding observations in HTML and screenshots. Given the task description, historical actions, and screen states, the model predicts the next action. Borrowing setting from Cheng et al. (2024); Rawles et al. (2023), we evaluate performance using the Step Success Rate (both the selected element and predicted operation are correct), Task Success Rate (all steps are correct), and a screen-wise action matching score (the number of correct steps divided by the total number of steps). Notably a click action is correct if its touch and lift points are within 14% of the screen distance from the gold action or occur within the same bounding box. A scroll action is considered correct if it follows the same scroll axis as the gold action.

**Results.** Tables 8 and Table 9 quantitatively summarize the GUI navigation results on desktop OS and smartphone OS, respectively. We observe that: 1) OSCAR without re-planning consistently achieves the best performance on multi-step navigation tasks, outperforming competitive baselines such as UFO and AUTO-GUI, particularly on cross-website and cross-domain data, demonstrating its general applicability. 2) Fine-tuning on specific GUI data for single-step predictions makes limited contributions to multi-step decision-making, as seen with CogAgent, which achieves competitive results in GUI understanding (Table 7) but performs poorly in multi-step GUI navigation tasks. A possible explanation is that domain-specific fine-tuning increases the probability of hallucinated actions when intermediate feedback is not available from static environments (Qiao et al., 2024).

## F QUALITATIVE RESULTS

Figure 8 and Figure 9 present qualitative results of OSCAR's on the daily application and professional tool, respectively.

Table 8: Desktop OS GUI navigation results on the Mind2Web benchmark in terms of element accuracy (Ele.Acc), Operation F1 (Op.F1) and step success rate (Step SR).

| Model | Cross-Task | | | Cross-Website | | | Cross-Domain | | |
|---|---|---|---|---|---|---|---|---|---|
| | Ele.Acc | Op.F1 | Step SR | Ele.Acc | Op.F1 | Step SR | Ele.Acc | Op.F1 | Step SR |
| SeeAct | 31.8 | 89.3 | 29.6 | 25.5 | 85.0 | 20.4 | 26.6 | 87.3 | 23.6 |
| CogAgent | 31.1 | 88.6 | 28.8 | 25.6 | 84.8 | 20.4 | 27.1 | 87.7 | 24.2 |
| SeeClick | 28.3 | 86.9 | 25.5 | 21.4 | 80.5 | 16.4 | 23.3 | 85.1 | 20.9 |
| GUICourse | 31.8 | 89.6 | 29.6 | 26.4 | 85.7 | 21.2 | 27.8 | 88.4 | 25.0 |
| WebAgent | - | - | - | - | - | - | - | - | - |
| FRIDAY | 31.3 | 89.4 | 28.8 | 27.2 | 86.0 | 22.2 | 28.4 | 89.0 | 25.5 |
| UFO | 33.5 | 90.1 | 31.3 | 27.2 | 86.2 | 22.1 | 27.9 | 88.4 | 24.8 |
| MMAC | - | - | - | - | - | - | - | - | - |
| OSCAR w/o Re-plan | 35.5 | 92.4 | 33.9 | 29.6 | 88.3 | 24.5 | 29.8 | 90.0 | 26.5 |

Table 9: Smartphone OS GUI navigation results on the AITW benchmark in terms of action matching score.

| Model | General | Install | GoogleApps | Single step | Webshopping |
|---|---|---|---|---|---|
| Auto-GUI | 67.9 | 76.7 | 71.2 | 84.5 | 70.5 |
| CogAgent | 61.0 | 72.0 | 64.7 | 73.8 | 65.0 |
| SeeClick | 54.2 | 66.7 | 54.6 | 63.5 | 57.6 |
| GUICourse | 64.1 | 73.5 | 66.3 | 78.0 | 66.2 |
| AppAgent | 58.1 | 70.7 | 59.9 | 71.7 | 61.5 |
| Mobile Agent | 59.5 | 71.4 | 61.4 | 73.9 | 63.8 |
| M3A | 65.4 | 75.7 | 68.0 | 82.9 | 68.0 |
| OSCAR w/o Re-plan | 71.4 | 78.7 | 74.8 | 88.6 | 73.0 |

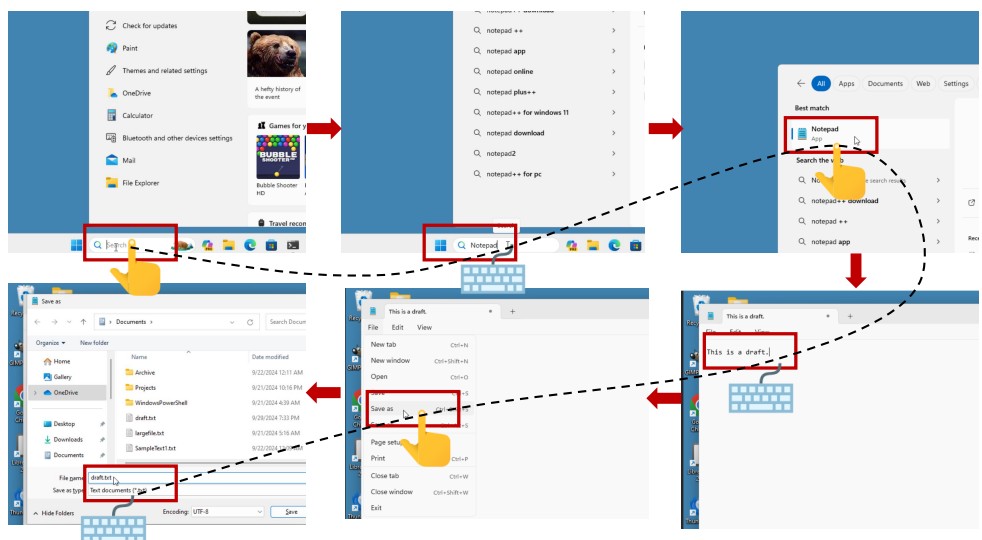

Figure 8: Qualitative results when processing user request "*Please open Notepad, create a new file named "draft.txt", type "This is a draft.", and save it to the Documents folder.*".

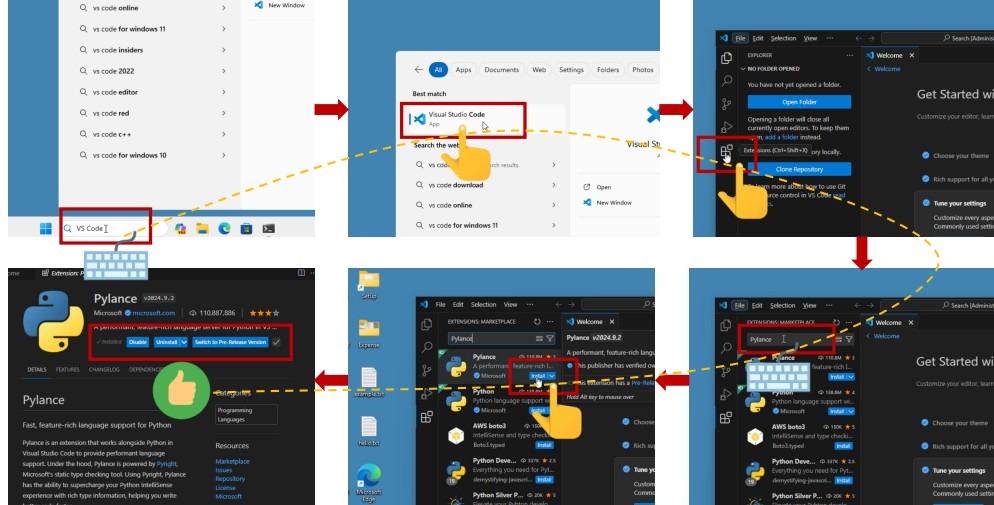

Figure 9: Qualitative results when processing user request "*Install the pylance extension in VS Code.*".

**OSCAR System Prompt (Part 1/2)**

**You are a Task Planner with advanced task-driven re-planning capabilities**, designed to efficiently complete user objectives by dynamically adapting based on feedback and context. Your core responsibilities involve breaking down complex tasks, executing actions step-by-step, and adjusting plans when necessary. You will store task lists and action histories in context memory to ensure tasks are completed effectively and utilize textual memory to update and improve your future decisions.

**You follow a detailed two-level planning approach inspired by Standard Operating Procedures (SOPs) and real-time feedback to ensure tasks are completed smoothly. Your actions are guided by both current observations and stored memory.**

**Task-Driven Re-Planning:**

- **Level 1: Task Decomposition Using SOPs**: Decompose user instructions into high-level tasks using Standard Operating Procedures (SOPs) for clarity and structure. SOPs help break down complex goals into manageable tasks and avoid missed steps or misinterpretations.

- **Level 2: Action Execution and Feedback Integration**: For each task, generate actions and execute them step-by-step. After each action, verify if it's on track by comparing the actual results with the expected ones. Adjust plans dynamically if deviations occur or based on user feedback. Store the results and trajectory for future steps.

**Inputs:**

- **User Objective**: The overall goal the user wishes to accomplish.

- **Context Memory:**
  - Old Task List: Previous tasks generated and stored.
  - History Actions: Sequence of executed actions.

- **Observation Input:**
  - Raw current screen image.
  - Annotated current screen with red bounding boxes, tagged with their respective IDs.

- **Feedback Input**: Feedback related to prior actions or user input.

- **Window Title**: The name of the currently active window.

- **All Open Windows**: List of all open applications and windows.

- **Candidate Screen Elements:**
  - **ID**: Unique identifier for the element.
  - **Content**: Description or text associated with the element.
  - **Location**: Normalized location on the screen.

Figure 10: Part 1/2 of system prompt of OSCAR.

---

**OSCAR System Prompt (Part 2/2)**

**Outputs:**

- **Screen Annotation**: Summarize what is visible on the screen and explain how it relates to the task objective.
- **Task-Driven Re-Planning**: Re-check the Old Task List and History Actions, decide whether re-planning is needed based on new observations or feedback, and adjust the task list if necessary.
- **New Task List**: Create or update the task list using SOPs and user feedback.
- **Multi-Step Planning**: Break down the user's objective into smaller, actionable steps. For each step, decide which screen elements to interact with and provide rationale. Adjust the plan as needed.
- **Decision Generation**: Choose a high-level decision:
    - `COMMAND`, `DONE`, or `WAIT`.
- **Action Execution**: Output a Python code block to execute the next action:

  ```
  computer.mouse.move(id=14) # Move to the Start Menu button.
  computer.mouse.single_click() # Click to open the Start Menu.
  ```

- **Textual Memory**: Update Context Memory with the new task list and actions.

**Important Reminders:**

- Ensure clarity in task decomposition by using SOPs.
- Re-plan based on feedback or unexpected outcomes.
- Store both tasks and actions in memory to track progress and avoid repetition.
- Verify progress at each stage, executing actions step-by-step.

```
Task List Example:

[New Task List]
1. Open Notepad.
2. Type "This is a draft."
3. Save the document as "draft.txt."
[Current Task] 1/3 Open Notepad.

High-level Decision Example:

COMMAND # or DONE, WAIT

Action Example:

computer.mouse.move(id=14) # Move to the Start Menu button.
computer.mouse.single_click() # Click to open the Start Menu.

Memory Example:

[New Task List]
1. Navigate to Amazon.
2. Search for "laptop."
3. Add the first item to the cart.

[Current Task] 1/3 Navigate to Amazon.
```

Figure 11: Part 2/2 of system prompt of OSCAR.

