# OpenReview forum: "OSCAR: Operating System Control via State-Aware Reasoning and Re-Planning"
_ICLR.cc/2025/Conference — ICLR 2025 Poster_

### Official Review · Reviewer_CNCy · 2024-10-27

**Soundness:** 2
**Presentation:** 2
**Contribution:** 2
**Rating:** 6
**Confidence:** 4

**Summary:**

This paper introduces OSCAR, a computer agent framework mainly focused on planning and aimed to generalize across diverse applications. The experiments demonstrate OSCAR’s effectiveness on GAIA, OSWorld, and AndroidWorld.

**Strengths:**

1. The proposed agent framework is formulated as a state machine, making it clear to follow.
2. OSCAR outperforms several baselines in both desktop and mobile environments.

**Weaknesses:**

1. The proposed modules (task-driven re-planning, context memory, GUI observations, and code actions) are not novel. Many closely related methods are not discussed, e.g.:
- Kim et al. "Language models can solve computer tasks." NeurIPS 2023.
- Sun et al. "Adaplanner: Adaptive planning from feedback with language models." NeurIPS 2023.
- Zheng et al. "Synapse: Trajectory-as-exemplar prompting with memory for computer control." ICLR 2024.

2. The experiments only ablate GUI observations and the replanning module. The analysis claims that the improvements come from “fewer re-planning attempts” and “smaller, more efficient steps”, but it is unclear how the proposed planning module contributes to this compared to the baselines.

3. In Table 3, OSCAR outperforms baselines in medium and hard cases but is worse in easy cases, for which the authors should analyze the potential reasons.

4. It is unclear how OSCAR leverages “real-time” feedback from the OS. Additional experiments on AgentStudio, a more complex, dynamic environment, are helpful.

5. The limitations of OSCAR are not discussed.

6. Minor: there are many typos, e.g.:
- line 300, “i.e. when” -> “i.e., when”; line 332, “(e.g. Chrome)” -> “(e.g., Chrome)”; and so on.
- missing or abundant space in Section 1 “(LLMs)(Ouyang …”, “(LMMs)(Li …”, Section 3 baselines “UFO(Zhang et al., 2024a)”, Table 1 caption “(i.e. GPT-4-turbo )”, etc.

I will raise my score if the authors resolve the above concerns.

**Questions:**

See Weaknesses above.

---

> ### Author Response · Authors · 2024-11-21
> **Author Response 1/3**
>
> > **R4-Q1**: The proposed modules (task-driven re-planning, context memory, GUI observations, and code actions) are not novel. Many closely related methods are not discussed, e.g.:
>
> > [1] Kim et al. "Language models can solve computer tasks." NeurIPS 2023.
>
> > [2] Sun et al. "Adaplanner: Adaptive planning from feedback with language models." NeurIPS 2023.
>
> > [3] Zheng et al. "Synapse: Trajectory-as-exemplar prompting with memory for computer control." ICLR 2024.
>
> **R4-A1**: Thank the reviewer for highlighting relevant works. While incorporating planning, memory, and feedback loops is indeed widely explored in agent design, we emphasize OSCAR's tailored framework for navigating dynamic OS environments, which introduces several key distinctions from both the baselines in Section 3 and the related works the reviewer mentioned.
>
> **Compared to the baselines in Section 3, the primary novelty of OSCAR lies in how it utilizes verification feedback for fine-grained task decomposition and re-planning**. FRIDAY uses directed acyclic graph-based planning, while UFO adopts chain-of-thought (CoT) planning; both generate detailed action steps directly from the input and must re-plan entirely from scratch upon negative feedback. In contrast, OSCAR’s hierarchical planning decomposes tasks into subtasks first and then formulates action steps. Combining with the state machine of OSCAR, this approach allows targeted adjustments during re-planning, making it more efficient and effective. As demonstrated empirically in Section 3, OSCAR’s strategy achieves notable improvements in task success rates and re-planning efficiency.
>
> **Compared to the works [1], [2], and [3] mentioned by the reviewer, OSCAR addresses fundamentally different challenges** due to its focus on real operating system environments (OSWorld, AndroidWorld) rather than their focus on MiniWob++ or ALFWorld, which are text-based simulation environments designed for web navigation or embodied reasoning. While MiniWob++ and ALFWorld are dynamic and provide real-time feedback—unlike the static environments such as Min2Web or AITW discussed in Appendix E of the original submission—the key differences between these environments and OSWorld lie in the complexity of both the observation and action spaces, which demand more comprehensive reasoning and planning capabilities from the agent.
>
> **For the observation space**, taking the MiniWob++ as an example, its environments rely on text-based HTML inputs, which are rendered visually but accessed directly through HTML parsing. This approach allows agents to interact using precise element locators, such as XPath queries, eliminating the need for visual understanding. In contrast, OSWorld and AndroidWorld require multimodal capabilities, as OSCAR relies on direct screen observations. Although OSCAR uses system APIs to retrieve accessibility trees for constructing GUI-grounded observations, this is a system-level primitive access rather than application-specific API calls. For instance, when navigating a browser within OSWorld, OSCAR does not parse HTML but instead interacts with the screen only, mirroring real-world human behavior more closely.
>
> **For the action space**, MiniWob++ test tasks are isolated within 53 independent environments (ALFWorld within 134), each designed with task-specific JavaScript scripts, such as `click-link` or `click-option`. Actions in MiniWob++ rely on XPath-based commands like `clickxpath //*[@id="subbtn"]`. This setup simplifies navigation through text-only operations but diverges significantly from real-world OS interactions. In contrast, OSWorld requires dynamic screen interactions, such as typing, clicking, and navigating between applications, presenting a substantially more complex and realistic control space.
>
> Overall, MiniWob++ agents achieve near-human success rates (e.g., [1]: 94%, [2]: 92%, [3]: 99%, vs. 93% human baseline reported by [3]). OSCAR, operating in a far more complex environment, achieves a state-of-the-art success rate of 24.5% on OSWorld, compared to a human performance of 72.36%. This underscores the challenges of navigating real OS environments and the need for innovations like OSCAR's approach.
>
> In summary, OSCAR's novelty lies in its **hierarchical planning** framework tailored specifically for real-world operating system environments, its **multimodal observation** that emulates human-like screen interactions, and its **integration of verification feedback into state machine control** for efficient and targeted re-planning. Unlike agents designed for MiniWob++ or ALFWorld, OSCAR bridges the gap between simulated and real-world computer control tasks, setting the stage for advancements in dynamic and complex agent environments.

---

> ### Author Response · Authors · 2024-11-21
> **Author Response 2/3**
>
> > **R4-Q2**: The experiments only ablate GUI observations and the replanning module. The analysis claims that the improvements come from “fewer re-planning attempts” and “smaller, more efficient steps”, but it is unclear how the proposed planning module contributes to this compared to the baselines.
>
> **R4-A2**: We appreciate the reviewer’s insights and suggestions. As stated in Line 428 of the original submission, the claims regarding “fewer re-planning attempts” and “smaller, more efficient steps” are based on direct experimental comparisons between OSCAR and the most competitive baseline, FRIDAY, which supports one of our core motivations outlined in Lines 94-98—the role of re-planning in navigation tasks.
>
> To further address the reviewer’s concerns, we performed an instance-level analysis on planning efficiency for ablation baselines employing different planning techniques, including Direct Prompt, ReAct, and Chain-of-Action, as detailed in Section 3.1. Following the same setup in Section 3.2, we analyzed the successful requests and tracked the number of re-planning occurrences before verification failures exceeded the maximum allowed attempts. The results are summarized in the table below.
>
> | Re-planning Attempts | 0 | 1 | 2 | 3 | 4 |
> | :----: | :----: | :----: | :----: | :----: | :----: |
> | Direct Prompt | 14% | 11% | 9% | 39% | 27% |
> | ReAct | 23% | 26% | 32% | 13% | 6% |
> | Chain-of-Action | 31% | 13% | 12% | 35% | 9% |
> | OSCAR | 33% | 24% | 30% | 10% | 3% |
>
> **The efficiency results reveal a slight inconsistency with the performance rankings** in Section 3.1. While performance rankings are as follows: Direct Prompt < ReAct < Chain-of-Action < OSCAR, efficiency rankings differ slightly: Direct Prompt < Chain-of-Action < ReAct < OSCAR. For example, like OSCAR, over 80% of samples using ReAct required fewer than three re-planning attempts, whereas for Chain-of-Action, only 56% met this threshold. Interestingly, 31% of samples in Chain-of-Action required no re-planning attempts, a higher ratio than ReAct.
>
> These findings suggest that while Chain-of-Action is a better one-trial planning method, it is less efficient in iterative re-planning processes, which involve refining plans based on feedback and historical actions. A potential explanation lies in the design of Chain-of-Action, which outputs a sequence of actions as intermediate histories and future plans without externalizing rationales step-by-step, as ReAct does. This limits its utility in leveraging context memory during re-planning.
>
> In contrast, OSCAR’s task-driven hierarchical planning—decomposing tasks before generating action plans—not only enhances the accuracy of initial plans but also facilitates efficient plan refinements during re-planning. This highlights the superiority of OSCAR in balancing performance and planning efficiency.
>
> > **R4-Q3**: In Table 3, OSCAR outperforms baselines in medium and hard cases but is worse in easy cases, for which the authors should analyze the potential reasons.
>
> **R4-A3**: As discussed in Lines 262-268 of the original submission, there are two main paradigms for designing agents to interact with dynamic environments for open-ended tasks: exploration-based and exploration-free methods. Exploration-based methods involve an initial phase where agents self-instruct tasks, comprehensively interact with the environment, and gather experiences in memory. During deployment, agents leverage the learned strategies to navigate new environments.
>
> AppAgent in Table 3 and FRIDAY in Table 2 exemplify exploration-based methods. While these methods can excel in tasks aligned with their exploration phase, they often face scalability challenges, particularly when environments are updated, such as the addition of new applications to an OS. This necessitates re-exploration to adapt, increasing inefficiency and the risk of overfitting to a limited exploration space. For instance, self-instructed tasks typically focus on simpler interactions, such as tasks involving fewer steps within a single application. This could explain why AppAgent outperforms OSCAR in easy cases but underperforms in medium and hard cases, where tasks demand more comprehensive reasoning and control capabilities.
>
> To address the challenges associated with self-instruct exploration—namely efficiency and task diversity—OSCAR adopts an on-the-fly re-planning strategy. By learning dynamically from experience and integrating verification feedback into its state machine control, OSCAR achieves state-of-the-art overall performance, particularly excelling in more complex task scenarios.

---

> ### Author Response · Authors · 2024-11-21
> **Author Response 3/3**
>
> > **R4-Q4**: It is unclear how OSCAR leverages “real-time” feedback from the OS. Additional experiments on AgentStudio, a more complex, dynamic environment, are helpful.
>
> **R4-A4**: We would like to emphasize that **both OSWorld and AndroidWorld are indeed dynamic environments with real-time feedback**, as also noted in the AgentStudio paper (see Table 7 in the arXiv v2 version released on Oct 3, 2024). The superior performance of OSCAR on these environments demonstrates its effectiveness.
>
> Regarding AgentStudio, during OSCAR's development and evaluation period (**before Oct 1, 2024**), it was in its beta stage with only 49 tasks across five software applications (e.g., FileSystem, Gmail, Google Docs, Google Calendar, VSCode), significantly fewer than the 369 tasks across nine applications in OSWorld. Thus, OSWorld served as a more comprehensive benchmark for our evaluation.
>
> By the ICLR discussion stage (**after Nov 13, 2024**), AgentStudio had been expanded to 205 tasks. While we acknowledge that including results on this benchmark could strengthen the analysis, we will attempt to conduct additional experiments during the discussion period. However, considering the complexity of configuring a new environment and the limited time available, this may pose challenges. If results are obtained in time, we will report them accordingly.
>
> > **R4-Q5**: The limitations of OSCAR are not discussed.
>
> **R4-A5**: Thank you for the valuable comment. The main limitations of OSCAR lie in two key aspects: **safety and scalability**.
>
> First, agent safety in realistic OS environments is critical. This includes system-intrinsic safety and misuse prevention. To address system-intrinsic safety, OSCAR employs virtualized environments (e.g., VMware/Docker for OSWorld, emulators for AndroidWorld) to isolate agent actions and prevent irreversible damage, significantly enhancing safety compared to related works like Cradle or FRIDAY, which develop agents directly in host environments. However, for real-world deployment without virtualized environments, the absence of reliable metrics for evaluating safety and detecting side effects remains a challenge. Currently, OSCAR’s [Verification] state focuses on task accuracy without fully accounting for harmful actions. Future work will explore mechanisms including action or tool filters for high-stakes interactions. Regarding misuse prevention, OSCAR's current capabilities, while state-of-the-art, are limited compared to human performance, reducing the risk of autonomous harmful actions, such as bypass CAPTCHA, misuse accounts, or exploit software vulnerabilities. Future iterations will integrate additional safeguards, such as requiring human confirmation for decisions with real-world consequences or tasks needing explicit consent.
>
> Second, the efficiency and scalability of OSCAR's state and memory representation require improvement. The current state machine employs pre-defined transitions, which are clear but lack structural semantics for enhanced adaptability. Future directions include integrating a dual-agent framework, with a supervisor agent autonomously managing state transitions, and leveraging structural representations like enumerated types or key-value pairs to enrich contextual information. For example, states like [Plan] and [Execute] could retain application-level actions to enable more flexible self-improvement. Additionally, advancing memory summarization and retrieval mechanisms could better support large-scale task deployments, addressing scalability challenges in increasingly complex environments.
>
> > **R4-Q6**: Minor: there are many typos, e.g.:
> line 300, “i.e. when” -> “i.e., when”; line 332, “(e.g. Chrome)” -> “(e.g., Chrome)”; and so on.
> missing or abundant space in Section 1 “(LLMs)(Ouyang …”, “(LMMs)(Li …”, Section 3 baselines “UFO(Zhang et al., 2024a)”, Table 1 caption “(i.e. GPT-4-turbo )”, etc.
>
> **R4-A6**: Thank you for pointing out these typographical issues. We will correct all mentioned typos throughout the paper.

---

> > ### Comment · Reviewer_CNCy · 2024-11-24
> >
> > Thanks for addressing my concerns. I've updated my rating.

---

### Official Review · Reviewer_1zyj · 2024-10-31

**Soundness:** 3
**Presentation:** 3
**Contribution:** 3
**Rating:** 8
**Confidence:** 3

**Summary:**

This work introduces OSCAR, a generalist agent that autonomously navigates and interacts with desktop and mobile applications through standard controls like mouse and keyboard inputs. OSCAR’s framework includes state transitions that dynamically adapt to various environments and are equipped with error-handling mechanisms. OSCAR also addresses the challenge of VLM’s difficulty in interpreting GUI screenshots by grounding them with semantic symbols. Finally, OSCAR incorporates task-driven replanning for efficient real-time adjustments based on feedback and exceptions. OSCAR is evaluated on GAIA, OSWorld, and AndroidWorld benchmarks. The results show that OSCAR significantly outperforms other baseline methods.

**Strengths:**

1. OSCAR demonstrates strong performance on evaluated benchmarks, achieving an average success rate of 28.7% and a success rate of 13.5% on the most challenging Level 3 tasks in the GAIA benchmark.
2. OSCAR implements a state machine with well-defined behaviors to manage complex, dynamic environments effectively.
3. Grounding GUI images with semantic symbols offers a more systematic approach for VLM to interpret the current state of the OS environment.

**Weaknesses:**

The [Verify] state is crucial to OSCAR’s framework; however, its exact functionality remains unclear. For instance, what occurs when [Verify] yields a false positive, such as validating an invalid plan? Can the system recover from these failures? Conducting an in-depth analysis of the [Verify] step's failure modes and success rate would significantly enhance the paper's quality.

**Questions:**

No questions

---

> ### Author Response · Authors · 2024-11-21
> **Author Response**
>
> > **R3-Q1**: The [Verify] state is crucial to OSCAR’s framework; however, its exact functionality remains unclear. For instance, what occurs when [Verify] yields a false positive, such as validating an invalid plan? Can the system recover from these failures? Conducting an in-depth analysis of the [Verify] step's failure modes and success rate would significantly enhance the paper's quality.
>
> **R3-A1**: Thank you for the reviewer’s thoughtful comments and suggestions. As noted in Lines 160–189 of the original submission, the [Verify] state plays a crucial role in OSCAR’s framework by providing feedback when task completion is indicated. In this state, OSCAR runs evaluation scripts to validate the outcomes of the executed actions. Unlike the feedback provided by the [Execute] state for individual interaction steps (e.g., clicking a button or typing text), the [Verify] state leverages task-specific evaluation scripts, which are provided by each benchmark.
>
> For example, on the OSWorld benchmark, an instruction like `Help me change the 2 in "H2O" to a subscript.` (0b17a146-2934-46c7-8727-73ff6b6483e8) is evaluated using the `compare_docx_files` function from the `evaluators\metrics\docs.py` script. This function compares the processed file against a gold standard file following a specific workflow, as outlined in the pseudocode:
>
> ```vbnet
> 1. Extract comparison options: ignore_blanks, ignore_case, ignore_order, content_only.
> 2. If files are invalid, return 0.
> 3. Load files:
>     - Use appropriate parser for .docx or .odt; return 0 on error.
>     - Extract paragraphs and sort if ignore_order is enabled.
> 4. If content_only:
>     - Normalize and compare entire text, applying ignore_case if needed.
>     - Return similarity score.
> 5. Else, normalize text if ignore_blanks is enabled.
>     - Compare paragraphs one by one; return 0 on mismatch.
> 6. Return 1 if all checks pass.
> ```
>
> For further details on this evaluation script, please refer to the OSWorld benchmark documentation. The script generates binary feedback: success or failure. Based on the outcome, OSCAR either transitions to the [Success] state if the verification passes or returns to the [Plan] state if it fails. If the failure exceeds the allowed maximum number of attempts, OSCAR transitions to the [Fail] state.
>
> When re-planning, OSCAR takes into account the history of plans and verification feedback to generate new plans for task decomposition. This task decomposition is autonomously determined by the model, and it involves revisiting previous actions. In many cases, the new plan includes undo operations (e.g., closing wrongly opened files or applications) to recover from failures.
>
> For the current version, we use publicly available implementations of the evaluation scripts without modifications to ensure a fair comparison across benchmarks. We acknowledge that more fine-grained verification and feedback during the [Verify] stage could improve re-planning efficiency. For example, providing structural feedback to each involved application could better guide plan refinement. This represents an exciting direction for future research and could help build more advanced benchmarks that reflect real-world scenarios more accurately.

---

### Official Review · Reviewer_Bipc · 2024-11-02

**Soundness:** 4
**Presentation:** 3
**Contribution:** 3
**Rating:** 6
**Confidence:** 4

**Summary:**

The paper presents OSCAR, a generalist agent designed to autonomously navigate and interact with both desktop and mobile applications using standard controls like mouse and keyboard. The agent aims to interpret user commands, interact with graphical user interfaces, and adapt its strategies based on real-time feedback. OSCAR is constructed as a state transition process and integrated with a GUI dual-grounding observation and task-driven re-planning. Further, the authors evaluate OSCAR on three digital task benchmarks, OSWorld, GAIA and AndroidWorld, outperforming current SOTA AI systems in completing digital tasks.

**Strengths:**

1. The use of a task-driven re-planning strategy allows OSCAR to adjust its actions based on real-time feedback, which enhances its ability to correct itself, do the state-aware reasoning and complete complex tasks autonomously.
2. OSCAR is evaluated against a diverse set of benchmarks, demonstrating superior performance in both desktop and smartphone environments, which underscores its generalizability and effectiveness.
3. The state transition process in OSCAR allows for systematic handling of tasks by structuring operations into distinct phases, such as observation, planning, execution, and verification. This structured approach enhances error recovery and adaptability by enabling the agent to assess and resolve issues at each state

**Weaknesses:**

1. I think it would be better to give some detailed cases to explain how the state transition works to complete the task and handle the errors.
2. There could be more discussions about how to fit OSCAR into different OS envs and generalize it to more benchmarks and systems.

**Questions:**

1. In the dual-grounding observation approach, how do you add the labels to each element to provide explicit semantic grounding?
2. What does "evaluation scripts" in section 2.1 refer to? Do you leverage the task-specific evaluation scripts from OSWorld? You may make it more clear about how you evaluate OSCAR on each of the three benchmarks.
3. As you allow 4 attempts per run for the agent to finish the task, is it crucial for a better performance and higher score on the benchmark? Do they always learn from failures and need more attempts to do better?
4. A more detailed error analysis is needed to specifically analyze the reasons for failure cases of reaching step limits, as this may help further improve OSCAR's error handling mechanism.

---

> ### Author Response · Authors · 2024-11-21
> **Author Response 1/3**
>
> > **R2-Q1**: I think it would be better to give some detailed cases to explain how the state transition works to complete the task and handle the errors.
>
> **R2-A1**: We appreciate the reviewer’s valuable suggestion and are currently updating the revised version of the paper with detailed cases illustrating state transitions and action outputs. These examples will be included in the qualitative results section in Appendix E to clarify how the state machine completes tasks and handles errors effectively.
>
> > **R2-Q2**: There could be more discussions about how to fit OSCAR into different OS envs and generalize it to more benchmarks and systems.
>
> **R2-A2**:
> OSCAR's core design, centered on the state machine and task-level re-planning, is inherently generalizable. However, adapting it to different benchmarks and operating systems (OS) primarily involves tailoring the interaction with the action space. Adapting OSCAR requires addressing two main aspects: **OS platforms** (e.g., desktop vs. smartphone) and **environment properties** (static vs. dynamic).
>
> For OS platform adaptation, as detailed in Table 5 of the submission, the main differences lie in the action space:
>
> **Desktop environments (e.g., OSWorld and GAIA)**: OSWorld relies on Python-based window parsing libraries such as pyatspi (Linux), pywinauto (Windows), or plistlib (macOS) to construct GUI-grounded observations. Interaction is controlled using pyautogui for mouse and keyboard actions. GAIA, on the other hand, leverages API-based tools like web search, audio-to-text, and image captioning.
>
> **Smartphone environments (e.g., AndroidWorld)**: These use touch and drag actions. AndroidWorld is built on AndroidEnv and Android Debug Bridge (ADB), which exposes a shell that allows users to send commands to the device emulator and manages interaction with the Android emulator, including obtaining screenshots and acceptability trees, and performing gestures and typing actions.
>
> It is important to note that all scripts used for adapting OSCAR to the benchmarks are based on publicly available implementations from the corresponding benchmark papers. No modifications were made to ensure fair comparisons.
>
> For environment property adaptation:
>
> **Dynamic environments (e.g., GAIA, OSWorld, and AndroidWorld)**: These provide real-time feedback and do not rely on gold reference action sequences, allowing multiple valid solutions for a given task. The focus here is on achieving task objectives rather than strictly following predefined action traces.
>
> **Static environments (e.g., Mind2Web and AITW in Appendix E)**: These benchmarks include high-level task descriptions, gold reference action sequences, and corresponding observations in HTML or screenshots. Agents are evaluated step-by-step against the gold reference, where a mismatch can result in incomplete task execution.
>
> Our experiments in Section 3, Appendix D, and Appendix E highlight OSCAR's effectiveness across diverse settings, including desktop and smartphone OS platforms, as well as dynamic and static environments. These results underscore OSCAR's generalizability across varying task types and feedback models.
>
> > **R2-Q3**: In the dual-grounding observation approach, how do you add the labels to each element to provide explicit semantic grounding?
>
> **R2-A3**: Adding descriptive labels and bounding boxes to screenshots for explicit semantic grounding involves three main steps:
>
> **Capture screenshot and extract accessibility tree**:
>
> A screenshot is obtained using `pyautogui`, and the accessibility tree is extracted by invoking the corresponding system API. This tree provides a structured representation of the UI elements.
>
> **Parse the accessibility tree into interaction elements**:
>
> The XML-format accessibility tree is parsed into a list of interaction elements, preserving key attributes such as `name`, `position`, and `size` from the original XML nodes. Based on these attributes, bounding box representations for each interaction element are constructed in the format `x`, `y`, `width`, `height`. Here, `x` and `y` represent the normalized coordinates of the top-left corner of the bounding box in the original screenshot, while width and height indicate the size of the bounding box.
>
> **Overlay bounding boxes and labels on the screenshot**:
>
> Using the bounding box coordinates and sizes, the list of interaction elements is drawn onto the screenshot. This is implemented using the Python Imaging Library (PIL), specifically leveraging `ImageDraw.Draw` to overlay rectangles and descriptive labels on the screen image.
>
> This process effectively provides explicit semantic grounding by visually annotating UI elements on the screenshot with meaningful labels and bounding boxes.

---

> ### Author Response · Authors · 2024-11-21
> **Author Response 2/3**
>
> > **R2-Q4**: What does "evaluation scripts" in section 2.1 refer to? Do you leverage the task-specific evaluation scripts from OSWorld? You may make it more clear about how you evaluate OSCAR on each of the three benchmarks.
>
> **R2-A4**: As mentioned in Line 335 of our submission, the "evaluation scripts" are used to verify modified files or displayed text content in windows, which are task-specific provided by each benchmark. We use them without modifications to ensure fair comparisons.
>
> For example, on the OSWorld benchmark, an instruction like `Help me change the 2 in "H2O" to a subscript.` (0b17a146-2934-46c7-8727-73ff6b6483e8) is evaluated using the `compare_docx_files` function from the `evaluators\metrics\docs.py` script. This function compares the processed file against a gold standard file following a specific workflow, as outlined in the pseudocode:
>
> ```vbnet
> 1. Extract comparison options: ignore_blanks, ignore_case, ignore_order, content_only.
> 2. If files are invalid, return 0.
> 3. Load files:
>     - Use appropriate parser for .docx or .odt; return 0 on error.
>     - Extract paragraphs and sort if ignore_order is enabled.
> 4. If content_only:
>     - Normalize and compare entire text, applying ignore_case if needed.
>     - Return similarity score.
> 5. Else, normalize text if ignore_blanks is enabled.
>     - Compare paragraphs one by one; return 0 on mismatch.
> 6. Return 1 if all checks pass.
> ```
>
> For more details on this evaluation script, please refer to the OSWorld benchmark.
>
> On the AndroidWorld benchmark, tasks like `Reply to the most recent text message using Simple SMS Messenger with: {message}` are evaluated using the `TestSimpleSmsReplyMostRecent` class found in the `task_evals/single/sms_test.py` script. This class verifies successful task execution by checking whether the expected message exists in the application. The evaluation process includes confirming the presence of both the correct phone number and the intended message body.
>
> For the GAIA benchmark, tasks follow a question-answering (QA) format, where the input may include files such as images or spreadsheets. The task output can take the form of strings, numbers, or lists, with each question having a single correct answer. This enables a quasi-exact match evaluation methodology.

---

> ### Author Response · Authors · 2024-11-21
> **Author Response 3/3**
>
> > **R2-Q5**: As you allow 4 attempts per run for the agent to finish the task, is it crucial for a better performance and higher score on the benchmark? Do they always learn from failures and need more attempts to do better?
>
> > **R2-Q6**: A more detailed error analysis is needed to specifically analyze the reasons for failure cases of reaching step limits, as this may help further improve OSCAR's error handling mechanism.
>
> **R2-A5** and **R2-A6**: The results in Section 3.2 of the original submission indicate that, among the tasks that are ultimately successful, only approximately 30% are accomplished on the first attempt, without requiring any re-planning. This highlights the importance of re-planning for achieving better performance, both for exploration-based agents like FRIDAY and for our exploration-free OSCAR agent. As noted in Lines 94–101, the need for re-planning is a key motivation for our work, as it enables more efficient interaction with dynamic environments. OSCAR’s state machine and task-driven re-planning mechanism allow for efficient re-planning, and over 80% of successful requests with OSCAR required fewer than 3 re-planning attempts. In comparison, for FRIDAY, more than 50% of successful tasks required 3 to 4 re-planning attempts. This shows that OSCAR’s approach is more efficient, requiring fewer re-planning attempts to achieve successful outcomes.
>
> To further evaluate the impact of re-planning on performance, we extended the maximum allowed attempts beyond four to explore the upper-bound performance without considering efficiency in terms of elapsed real time or input context window. It is important to note that some tasks remain unaccomplished even with unlimited attempts. The results, expressed as the success ratio of tasks completed within different numbers of attempts relative to the entire task set, are summarized in the table below:
>
> | Re-planning attempts |   $\le$ 4   | $\le$ 8 | $\le$ 12 | $\le$ 16 | $\le$ 20 | $\gt$ 20 |
> | :----------: | :----------: | :----------: | :----------: | :----------: | :----------: | :----------: |
> | OSCAR | 24.5 | 28.2 | 28.2 | 28.2 | 28.2 | 28.2 |
> | FRIDAY | 18.8 | 20.6 | 21.6 | 22.5 | 23.3| 23.3 |
>
>
> Among the 369 tasks in OSWorld, OSCAR achieves no additional task completions beyond 8 attempts, indicating its performance plateau. In contrast, FRIDAY continues to improve until 20 attempts before reaching its plateau.
> Besides, while additional re-planning attempts enhance performance for FRIDAY, the improvements become marginal after 8 attempts, with only a 1% increase in success rate (equivalent to solving 1–2 additional tasks out of 369).
> Overall, OSCAR attains an upper-bound performance of 28.2%, compared to 23.3% for FRIDAY. This performance gap likely arises from the inherent differences in their planning techniques: OSCAR employs a two-level task-driven planning mechanism, whereas FRIDAY uses directed acyclic graph-based planning.
> Additionally, we performed a failure case analysis on OSCAR for tasks that could not be completed even with unlimited attempts. The reasons for task failure within the step limit fall into three primary categories:
>
> **45%: Wrong planning** – Examples include failing to close pop-ups on real-world web pages or being distracted by advertisement content.
>
> **29%: Incorrect action execution** – Issues such as mouse click inaccuracies frequently result in repetitive misclicks during execution.
>
> **26%: Faulty GUI understanding and grounding** – This includes challenges like navigating professional software using keyboard shortcuts, misreferencing GUI elements in generated Python code, and triggering interpreter errors during state transitions (e.g., from Plan → Execute or Execute → Plan), which lead to repetitive, non-productive actions.
>
> These findings underscore the need for future work to focus on improving GUI understanding and grounding, as well as incorporating prior knowledge to better handle environmental noise. such as robustly addressing advertisement distractions and implementing mechanisms to recover from page errors or unexpected states.

---

> > ### Comment · Reviewer_Bipc · 2024-11-22
> >
> > Thanks for your reply! I am more inclined to think that this paper should be accepted.

---

### Official Review · Reviewer_TSTo · 2024-11-08

**Soundness:** 3
**Presentation:** 4
**Contribution:** 3
**Rating:** 8
**Confidence:** 3

**Summary:**

- This work presents an LLM+LMM-based agent to solve Desktop/Mobile OS tasks, including key modules such as GUI elements recognition and SoM prompting, plan/re-plan state machine and code execution for general mouse/keyboard control.
- Experiments on multiple, both static and dynamic, benchmarks demonstrate the proposed framework's efficacy and generalizability.

**Strengths:**

- This work shows a robust and efficient OS agent by resorting to a general purposed language models without fine-tuning. To cope with GUI recognition difficulties (beyond general vision tasks), SoM techniques are applied to improve reliability.
- Many baseline comparisons and detailed implementation are also provided.

**Weaknesses:**

- Discussion on safety may be helpful since the agent has control over an OS.
- Although it uses a state machine and re-planning, OSCAR lacks a self-improvement mechanism. Do you have more insights on representing states, e.g. system errors/task failures, beyond pure text language?

**Questions:**

- What makes OSCAR outperform other baselines mostly (since many methods also have their feedback loop)?
- Will a fine-tuned LM be eventually required for the robustness in real-world scenarios beyond those benchmarks? To what degree is a user/expert required to supervise the agent's operation?
- Is plan -> execution -> error -> re-plan enough even for complex tasks or sampling-based methods such as evolutionary search (but costly and redundant) should be also considered.

---

> ### Author Response · Authors · 2024-11-21
> **Author Response 1/3**
>
> > **R1-Q1**: Discussion on safety may be helpful since the agent has control over an OS.
>
> **R1-A1**:  We agree with the reviewer that agent safety, particularly when interacting with an OS, is a critical issue. This encompasses two primary aspects: **system-intrinsic safety** and **misuse prevention**.
>
> For system-intrinsic safety, we emphasize transparency and protection during development to mitigate unintended consequences. Specifically, we employ virtualized environments, such as VMware/Docker in OSWorld and emulators in AndroidWorld benchmarks, to isolate agent actions and prevent irreversible damage to host machines. This approach significantly enhances safety compared to the related works like Cradle, FRIDAY, or OS-Copilot, which develop agents directly in host environments. However, for real-world deployment, the lack of reliable metrics to evaluate agent safety and detect latent side effects remains a challenge. Currently, our [Verification] state primarily assesses task accuracy but does not fully account for harmful actions. We recognize the importance of improving safety alignment, particularly in human-computer interactions, and plan to explore mechanisms like action or tool filters for critical actions.
>
> Regarding misuse prevention, agents operating in realistic environments could theoretically bypass CAPTCHA, misuse accounts, or exploit software vulnerabilities. In our benchmark evaluations, however, OSCAR—despite achieving state-of-the-art performance—still exhibits a significant gap compared to human proficiency (24.5% for OSCAR vs. 72.36% for humans in OSWorld). This limited capability prevents OSCAR from functioning as a fully generalist agent and ensures it follows task instructions without autonomous harmful behavior. As more advanced models emerge, we aim to integrate additional safeguards into OSCAR’s state machine to mitigate potential risks and enhance safety, for example, ask a human to confirm decision that may results in meaningful real-world consequence as well as any tasks requiring affirmative consent, such as accepting cookies or agreeing to terms of service.
>
> > **R1-Q2**: Although it uses a state machine and re-planning, OSCAR lacks a self-improvement mechanism. Do you have more insights on representing states, e.g. system errors/task failures, beyond pure text language?
>
> **R1-A2**: We sincerely appreciate the reviewer’s insightful and thought-provoking comments.
>
> First, we would like to emphasize that the state machine and re-planning mechanisms inherently support a form of self-improvement, specifically cross-trial improvement within the current task/instruction. OSCAR leverages verification feedback from previous trials to optimize fine-grained task re-planning. Empirical results in Section 3.2 highlight this efficiency, showing that OSCAR requires fewer re-planning attempts and that these re-plans involve smaller, more targeted steps.
>
> For cross-task/instruction self-improvement, which involves enabling the agent to learn from web resources or prior experiences, we acknowledge its potential. For example, an agent could use a failed trial to search for better plans online or store successful action plans in an external procedural memory to handle similar future tasks more effectively. However, as discussed in Lines 268–290 of the paper, such cross-task strategies often introduce scalability challenges and inefficiencies, particularly when managing a large task volume. These methods may also risk overfitting to a limited exploration space. For instance, self-instructed tasks typically focus on simpler interactions, such as tasks involving fewer steps within a single application. Thus, in this work, we prioritize dynamic interaction with OS environments through the state machine, achieving superior performance compared to cross-task self-improvement baselines like FRIDAY.
>
> Lastly, we recognize the potential of advanced self-improvement strategies, particularly when integrated with structural state representations in our state machine. Currently, the state machine uses pre-defined transitions, which are atomic and clear but lack structural semantics. A more flexible approach, such as a dual-agent framework with a supervisor agent autonomously managing state transitions, could significantly enhance adaptability. Additionally, incorporating structural state representations—such as enumerated types or key-value pairs beyond plain text—would provide richer contextual information. For instance, the [Plan] and [Execute] states could record actions associated with each active application, and maintaining these records could enable more flexible application-level self-improvement. Furthermore, we identify opportunities to explore efficient memory summarization and retrieval mechanisms to better support large-scale task deployments. These directions represent promising avenues for future research.

---

> ### Author Response · Authors · 2024-11-21
> **Author Response 2/3**
>
> > **R1-Q3**: What makes OSCAR outperform other baselines mostly (since many methods also have their feedback loop)?
>
> **R1-A3**: As shown in our ablation results in Section 3.1, OSCAR’s superior performance is primarily due to its GUI-grounding and task-driven re-planning capabilities.
>
> While several baselines include feedback loops, the key distinction is how OSCAR leverages verification feedback to inform its hierarchical planning. For example, FRIDAY uses directed acyclic graph-based planning, and UFO applies chain-of-thought (CoT) planning; both generate detailed action steps directly from input and, upon receiving negative feedback, must re-plan from scratch by rewriting all action steps. In contrast, OSCAR’s hierarchical planning first decomposes tasks into subtasks before formulating action steps, allowing more targeted re-planning. This fine-grained approach enables OSCAR to make more efficient and effective adjustments based on feedback, as also demonstrated empirically in Section 3.2.
>
> > **R1-Q4**: Will a fine-tuned LM be eventually required for the robustness in real-world scenarios beyond those benchmarks? To what degree is a user/expert required to supervise the agent's operation?
>
> **R1-A4**: Given the open-ended nature and broad scope of computer control tasks, the effectiveness and robustness of fine-tuning vary across scenarios. Specifically, in our GUI static navigation experiments in Appendix E of original submission, fine-tuned agents like CogAgent (fine-tuned CogVLM-17B) and GUICourse (fine-tuned Qwen-VL-9.6B) exhibit comparable desktop OS navigation performance to GPT-4V-based SeeAct, as shown in Table 8. However, in smartphone OS navigation tasks (Table 9), both CogAgent and GUICourse significantly underperform SeeAct, despite their fine-tuning datasets including smartphone UI data, OCR grounding, and GUI knowledge. These results suggest that while fine-tuning can be effective for domain-specific applications, it faces challenges in achieving robust cross-domain and cross-environment generalization.
>
> Beyond fine-tuning, external agent designs can also enhance effectiveness and robustness in real-world scenarios. For instance, OSCAR leverages task-driven re-planning, FRIDAY employs exploration-based self-improvement, and Cradle incorporates skill summarization. These designs provide complementary approaches to address the limitations of fine-tuning alone.
>
> Regarding user supervision, as discussed on the safeguard in **R1-A1**, our agent currently operates autonomously in most cases, requiring minimal user intervention only during error-handling states (e.g., managing excessive processes or addressing model backend crashes). However, in real-world applications, some degree of expert oversight may remain necessary, particularly for novel or high-stakes tasks where safety is paramount. Looking ahead, we aim to incorporate adaptive learning mechanisms to further reduce reliance on user supervision while maintaining safety and task accuracy.

---

> ### Author Response · Authors · 2024-11-21
> **Author Response 3/3**
>
> > **R1-Q5**: Is plan -> execution -> error -> re-plan enough even for complex tasks or sampling-based methods such as evolutionary search (but costly and redundant) should be also considered.
>
> **R1-A5**: Current results in Table 2 and Figure 6 demonstrate that OSCAR's re-planning approach surpasses exploration-based methods, such as FRIDAY, in task performance and efficiency. To further explore the potential benefits of integrating exploration-based strategies into OSCAR's workflow, we conducted an ablation study augmenting OSCAR with an exploration-based approach.
>
> Specifically, we applied FRIDAY's exploration setting within the OSWorld benchmark. In this setup, the agent leverages 4 benchmark samples as in-context demonstrations to self-instruct a continuous stream of tasks. The tasks begin with simpler objectives, such as navigation within single applications, and gradually increase in complexity, requiring interactions across workflows involving multiple applications. Using the defined basic actions—mouse behaviors and keyboard interactions as outlined in Table 5—the agent employs trial and error to accumulate successful trials into reusable, self-generated actions (referred to as "self-generated tools" in FRIDAY's paper for the GAIA benchmark) implemented in Python code. This process enables the agent to adapt more efficiently over time. **Additionally, we evaluated generalization by excluding the "Prof. Software" from the exploration process**.
>
> The success rate, along with the number of self-instructed tasks and the elapsed real time (wall-clock time) spent during the exploration process, are summarized in the table below.
>
> | # Self-instructed Tasks | Wall-clock Time | | | Success Rate | | | |
> | :----: | :----: |   :----: |  :----: |  :----: |  :----: |  :----: |  :----: |
> |  |  | OS | Office | Daily | Prof. | Multi | Avg. |
> | 0 | 0 min | 58.3 | 12.0 | 16.7 | **22.4** | **12.9** | **24.5** |
> | 100 | 93.8 min | 58.3 | 12.0 | 16.7 | 22.4 | 12.9 | 24.5 |
> | 200 | 172.5 min | 58.3 | 12.0 | 16.7 | 22.4 | 12.9 | 24.5 |
> | 400 | 325.7 min | **62.5** | **12.8** | **19.2** | 14.3 | 11.9 | 24.3 |
>
> Notably, after sufficient exploration, incorporating exploration-based augmentation resulted in approximately a 10% performance improvement on tasks involving software explored during the exploration stage. However, it led to a notable 36% performance decrease on tasks involving un-explored software, such as the "Prof. software", and a slight decrease in performance on multi-application tasks, resulting in slightly worse results than the original OSCAR. These findings underscore a key limitation of exploration-based methods: achieving robust generalization remains a significant challenge.
>
> In summary, exploration-based strategies hold promise for enhancing agent performance when exploration is diverse and comprehensive, though this comes at the cost of time. Addressing the efficiency of such methods, as well as ensuring the diversity and depth of the self-instruction process, is critical for achieving scalability and applicability in real-world scenarios.

---

### Meta-Review · Area_Chair_CEHN · 2024-12-17

**Metareview:**

**Summary**: This paper presents OSCAR, a generalist agent designed for autonomous interaction with desktop and mobile operating systems. OSCAR uses state-aware reasoning and task-driven re-planning to dynamically interact with graphical user interfaces (GUIs) through mouse and keyboard inputs. The agent processes screen observations, translates user instructions into executable Python code, and adapts to errors using a state machine framework. OSCAR’s robustness and adaptability are validated through extensive experiments on benchmarks such as OSWorld, GAIA, and AndroidWorld, where it outperforms current state-of-the-art systems. The paper highlights OSCAR’s ability to generalize across dynamic OS environments and handle real-time feedback effectively.

**Strengths**:
- All reviewers praise the thorough and diverse evaluations across dynamic OS benchmarks, including OSWorld, GAIA, and AndroidWorld. OSCAR achieves state-of-the-art performance, demonstrating its ability to adapt to various tasks and outperform existing agents like FRIDAY and UFO.
- Reviewers recognize OSCAR’s dual-grounding approach for GUI understanding, combining visual and semantic grounding, as a significant advancement. This enables the agent to interpret complex user interfaces more accurately and robustly, enhancing task execution.

**Weaknesses**:
- Safety concerns: Reviewer `TSTo` raises concerns about safety, particularly when an agent has autonomous control over operating systems. The authors admit that safety (and scalability) issues are the biggest limitations of the paper, and provide more discussion in the rebuttal. While the authors discuss virtualization and safeguards, a more thorough exploration of potential misuse and real-world deployment risks would strengthen the paper.
- Scalability concerns: Reviewer `TSTo` questions whether OSCAR’s current plan-execute-replan cycle is sufficient for highly complex tasks. Exploration-based methods like evolutionary search could provide additional robustness but were not fully considered in this work.
- Generalization to other OSs: Reviewer `Bipc` points out that adapting OSCAR to other operating systems and benchmarks requires additional effort. While the authors clarify that the modular design aids generalizability, further examples or discussions would improve confidence in OSCAR’s scalability.
- Lack of self-improvement mechanisms and error analysis discussion: Reviewer `TSTo` notes the lack of advanced self-improvement mechanisms and reviewer `Bipc` requests more discussion on error analysis. While OSCAR handles task failures well through state-driven re-planning, it does not learn from cross-task experiences or explore structural improvements beyond text-based state representation.

**Recommendation**: Based on the reviewers' feedback and discussion, the paper offers an innovative, well-executed solution for dynamic OS control tasks. While safety considerations and generalization to new environments remain areas for improvement, the authors' rebuttal addressed these concerns adequately. In my opinion, OSCAR’s strong performance, technical contributions, and experimental validation make it a valuable addition to the field, and my vote is for Accept as a poster.

**Additional Comments On Reviewer Discussion:**

During the rebuttal period, the reviewers raised the following key points:
- Reviewer `TSTo` emphasized the need for more discussion on the safety implications of OSCAR, given its ability to control operating systems autonomously. In response, the authors clarified that OSCAR operates within virtualized environments (e.g., VMware, Docker, and Android emulators) to ensure safety during task execution. They also discussed plans for future safeguards, such as human confirmation for critical actions. This response addressed the reviewer's concerns about system-intrinsic risks and misuse prevention, although real-world deployment safety remains an open challenge. I would like the camera ready version of the manuscript to incorporate such a discussion.
- Reviewer `Bipc` questioned how well OSCAR can adapt to other operating systems and benchmarks. The authors explained that OSCAR’s modular design enables adaptability, with platform-specific scripts managing interactions for desktop and smartphone environments. They provided further details on how OSCAR handles action spaces and environment properties (e.g., static vs. dynamic environments), demonstrating its scalability. This clarification strengthened confidence in OSCAR’s ability to generalize across platforms. Once again, the manuscript would benefit from including such a discussion.
- Reviewer `TSTo` and `Bipc` noted the lack of self-improvement mechanisms and requested further insights into OSCAR’s error-handling processes. The authors explained that OSCAR already supports dynamic re-planning based on verification feedback and highlighted potential extensions, such as integrating structural state representations and cross-task learning for future work. These responses provided sufficient explanation for OSCAR’s current limitations while acknowledging areas for improvement.
- Reviewer `TSTo` questioned whether OSCAR’s current plan-execute-replan approach is sufficient for highly complex workflows. The authors conducted an ablation study exploring the integration of exploration-based methods (e.g., FRIDAY’s exploration setting). Results showed that exploration-based methods provided marginal improvements but reduced generalization across tasks. This study addressed the concern by demonstrating OSCAR’s efficiency and adaptability compared to alternatives.

As I see it, the authors' rebuttal was thorough and addressed the reviewers' concerns with clear explanations and additional experiments. While some points—such as safety and self-improvement—remain areas for future work, the reviewers acknowledged the authors' efforts and some of them increased their scores. These discussions reinforced my final decision to recommend acceptance, as the strengths of the paper significantly outweigh the remaining limitations.

---

### Decision · Program_Chairs · 2025-01-22

Accept (Poster)